

# Multistatic meteor radar observations of gravity wave-tidal interaction over Southern Australia

Andrew J. Spargo[1], Iain M. Reid[1,2], and Andrew D. MacKinnon[1]

[1]Department of Physics, School of Physical Sciences, The University of Adelaide, Adelaide, 5005, Australia
[2]ATRAD Pty. Ltd., 20 Phillips St., Thebarton, 5031, Australia

**Correspondence:** Andrew J. Spargo (andrew.spargo@adelaide.edu.au)

**Abstract.**

This paper assesses the ability of a recently-installed 55 MHz multistatic meteor radar to measure gravity wave-driven momentum fluxes around the mesopause, and applies it in a case study of measuring gravity wave forcing on the diurnal tide during a period following the autumnal equinox of 2018. The radar considered is in the vicinity of Adelaide, South Australia (34.9°S, 138.6°E) and consists of a monostatic radar and bistatic receiver separated by approximately 55 km.

The assessment shows that the inclusion of the bistatic receiver reduces the relative uncertainty of the momentum flux estimate from about 75% to 65% (for a flux magnitude of $\sim 20$ m$^2$s$^{-2}$, one day's worth of integration, and for a gravity wave field synthesized from a realistic spectral model). This increase in precision appears to be entirely attributable to the increased number of meteor detections associated with the combined monostatic and bistatic receivers, rather than changes in the meteors' spatial distribution.

The case study reveals large modulations in the diurnal tidal amplitudes, with a maximum tidal amplitude of $\sim 50$ ms$^{-1}$ and an associated maximum zonal wind velocity of around 140 ms$^{-1}$. While the observed gravity wave forcing exhibits a complex relationship with the tidal winds during this period, the components of the forcing are seen to be approximately out of phase with the tidal winds above 88 km. No clear phase relationship has been observed below 88 km.

## 1 Introduction

It has been known for over three decades that the momentum deposition arising from the dissipation of atmospheric gravity waves (herein GW forcing) has a major influence on the background wind and thermal structure of the mesosphere-lower-thermosphere/ionosphere (MLT/I; $\sim$ 80-100 km altitude) (Fritts, 1984). The small scales of the GWs relative to typical grid spacing in global climate models (GCMs) has led to a need to incorporate accurate parameterizations of the GW forcing within the GCMs (Kim et al., 2003; Ern et al., 2011). To support this need, there have been dozens of ground-based, satellite and in-situ studies of the associated GW momentum fluxes in the MLT/I (see e.g., Fritts et al. (2012a) and Nicolls et al. (2012) and references therein). Even so, many of the effects of GWs in the MLT/I are still acknowledged to be poorly understood, which continues to motivate major observational campaigns (e.g., Fritts et al. (2016)).



In recent years, monostatic meteor radars have been the most widely deployed of those ground-based instruments (e.g., Hocking (2005); Antonita et al. (2008); Clemesha and Batista (2008); Beldon and Mitchell (2009, 2010); Clemesha et al. (2009); Fritts et al. (2010a, b, 2012a, b); Vincent et al. (2010); Placke et al. (2011a, b, 2014, 2015); Andrioli et al. (2013a, b, 2015); Liu et al. (2013); de Wit et al. (2014b, a, 2016); Matsumoto et al. (2016); Riggin et al. (2016); Jia et al. (2018)). This is largely due to the low cost and ease of installing and continuously running meteor radars relative to other instruments capable of making the same measurements, such as partial reflection radars (e.g., Vincent and Reid (1983)), coherent radars (e.g., Reid et al. (2018b)), incoherent scatter radars (e.g., Nicolls et al. (2012)), and Doppler lidars (e.g., Agner and Liu (2015)).

Like all other ground-based radar observations of momentum fluxes (see e.g., the discussions in Fritts et al. (2012a), Spargo et al. (2017) and Reid et al. (2018b)), there are concerns around the accuracy and precision of the estimates derived from meteor radar. As shown by Vincent et al. (2010), the measurement uncertainties are dependent on both the meteor detection rates and the complexity of the GW spectrum. Their results showed that even at the altitude of the peak of the meteor distribution, integration times of the order of a month or longer may be needed to definitively estimate the sign of the flux, for typical flux magnitudes. Fritts et al. (2012a) and Andrioli et al. (2013a), who also incorporated real time and spatial meteor distributions and a wider variety of GW fields in their simulation, reach similar qualitative conclusions, although Fritts et al. (2012a) in particular argue that their measurement uncertainties for a composite day of data comprising measurements spanning one month may be much smaller than those reported in Vincent et al. (2010), due to the use of a larger total number of meteors and an assumption that the wave field in the MLT/I is often dominated by large amplitude monochromatic waves.

Given the demonstrated sensitivities of momentum flux estimation uncertainties, it is important that all users of meteor radars appreciate the uncertainties specific to their radar configuration (the count rates and count distribution, the radar location, and the time of year, and the likely GW field) prior to interpretation of their measurements. This study considers such a simulation of momentum flux measurement uncertainties from a 55 MHz meteor radar in a mid-latitude Southern Hemisphere (SH) site in Australia, and bears those uncertainties in mind in the interpretation of a case study of GW forcing on the diurnal tide. The aspects of this study that are unique can be summarized as follows:

- we consider a multistatic meteor radar configuration consisting of a monostatic radar and a bistatic receiver separated by $\sim 55$ km;

- we propagate realistic levels of receiver noise and mean phase bias to the angle-of-arrival (AOA) and radial velocity estimates, that are used in the subsequent momentum flux estimation;

- a realistic GW spectral model is used to synthesize the wind field from which the momentum fluxes arise.

Section 2 briefly overviews the radar configuration, the count rates obtained and the phase calibration offsets applied. Section 3 gives a detailed description of the simulation that estimates the momentum flux measurement uncertainties, and its results. Section 4 presents a case study of momentum fluxes estimated using the radar during the Austral winter, and attempts to validate them by looking at the interaction between the measured fluxes and the tidal winds. Discussion and conclusions follow.



**Table 1.** Experiment parameters used for the BP meteor radar transmitter, for all data presented in this paper.

| Parameter | Value |
| --- | --- |
| Frequency | 55 MHz |
| Pulse width | 7.2 km |
| Pulse code | 4-bit complementary |
| Pulse shape | Gaussian |
| PRF | 440 Hz |
| Range sampling | 68.4-309.6 km |
| Range sampling interval | 1.8 km |
| Peak power | 40 kW |
| Polarisation | Circular |

## 2 Instrumentation

The multistatic meteor radar considered in this study consists of a Stratosphere-Troposphere (ST)/Meteor radar located at the Buckland Park (BP) field site (34.6°S, 138.5°E) (briefly described by Reid et al. (2018a)), and a remote receiving system located near the township of Mylor, South Australia (35.1°S, 138.8°E) (about 55 km to the South-East of BP).

In meteor mode on the BP system, a single crossed folded dipole is used for transmission and a five-element interferometer arranged in a configuration identical to that of Jones et al. (1998) is used for reception. Three element Yagi antennas are used for the interferometer's receive antennas. A peak power of 40 kW is used on transmission. Other experimental parameters used are summarized in Table 1, and a detailed description of the radar hardware is given in Dolman et al. (2018).

The remote receiver system consists of a six receive channel digital transceiver identical to the transceiver system of the
BP ST/Meteor radar. In the current configuration, only five of those receive channels are used. The same five receiver antenna arrangement is used at the remote site. To permit accurate range and Doppler estimates at the remote site, the system timing, frequency and clocks at both sites are synchronized with GPS-disciplined oscillators (GPSDOs).

The techniques used to estimate various data products from the received meteor echoes, including radial velocity, meteor position, signal-to-noise ratio (SNR), and decay time, follow those outlined in Holdsworth et al. (2004).

The dataset considered spans 17th March 2018 to 9th September 2018, with few interruptions (the number of meteors detected per day on both receivers for this interval are shown in Fig. 1).

### 2.1 Receiver channel phase calibration

Compensating for any systematic receiver channel phase offsets plays an important role in ensuring the accuracy of the position and height estimates of the detected meteors. To calibrate the phases of the receive channels for both of the meteor receiver
interferometers used in this study, we have followed the approach suggested in Holdsworth (2005).



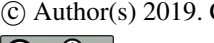

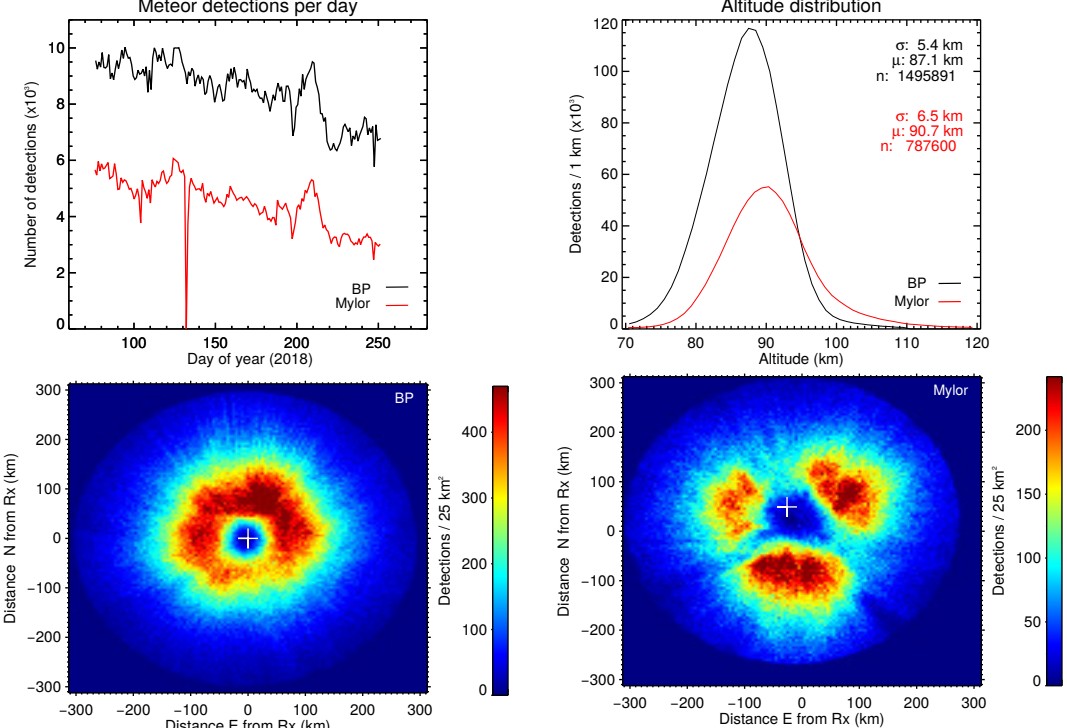

**Figure 1.** Upper row: the meteor detection rates for the BP and Mylor receivers over the 2018 campaign (left) and the associated distribution of meteors in altitude (right). Lower row: the horizontal distribution of meteors for the BP (left) and Mylor (right) receivers. The transmitter location in each case is denoted by a white cross.

The Holdsworth (2005) approach determines the offsets to apply to the phase differences between the centre and each of the other receive antenna channels that maximise the number of meteors within a range of heights that the meteors are expected to occur (see Chau and Clahsen (2019) for a generalized approach to this). For the BP system, we have used minimum and maximum permissible heights of 70 and 110 km, respectively, and 70 and 120 km for the Mylor system. A slightly larger

5    height interval has been used for the Mylor system to allow for the effect of the distribution of Bragg wavelengths on the meteor height distribution width (see, e.g., Stober and Chau (2015), Sect. 3 for a description of this effect).

The phase offsets applied to the Mylor system, and the variabililty of the offsets for the BP system (for which a fixed calibration was used) are shown in Fig. 2. We note a stable calibration for BP, but a few sudden "shifts" in the Mylor case; this has subsequently been determined to be due to a slight rotation of the antenna elements by local wildlife. We do not expect

10    isolated shifts like this to have an adverse impact on the analysis performed in this paper, although to somewhat compensate for it, we have performed a daily re-calibration of the receiver channels (using the calibration results for each day) before subsequent processing of the data.





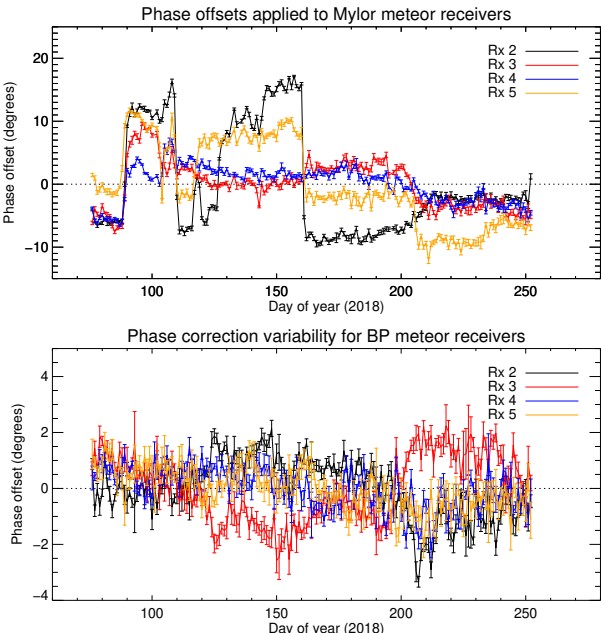

**Figure 2.** Phase offsets applied to Mylor meteor radar antennas as a function of time (top), and the phase offsets indicated by the Holdsworth (2005) calibration procedure on BP meteor radar data (bottom).

## 3 Simulation of wind covariance estimation

### 3.1 Simulation overview

The aim in developing this simulation has been to quantify the uncertainties in the $\langle u'w' \rangle$ and $\langle v'w' \rangle$ covariance components derived from meteor echoes received in an arbitrary network of meteor radar transmitters and receivers, and to be able to

5   characterize the dependence of those uncertainties on the network shape and the spectrum of the GWs constituting the input wind field. The basic workflow of the simulation (all components of which are elaborated upon in subsequent subsections) may be summarized as follows:

1. Produce a sample of meteors in space and time for each site under consideration, by sampling from realistic spatio-temporal meteor detections corresponding to each site.

10   2. Specify a wind field based on the superposition of monochromatic gravity waves derived from a realistic GW spectrum, and compute the wind velocities at each of the simulated meteors.

3. Compute the "radial" wind velocity measured at the receiver associated with each meteor detection.

4. For each meteor-site combination, synthesize in-phase and quadrature (IP and Q)-time series for each receiver at the site, based on the "radial" velocity and AOA of the meteor.





5.  Add a realistically-sized phase bias and noise floor to each receiver channel.

6.  Estimate the "radial" velocity and AOA of the meteor from the simulated time series.

7.  Estimate the wave field covariances using the meteors retrieved from different combinations of sites.

8.  Return to 1., and repeat for the number of realisations required to produce covariance error distributions (in the next step) of the desired statistical significance and resolution.

9.  Compare the estimated covariances with those computed directly from the 3D wind velocities at the meteors and those calculated at 2-minute resolution at the origin of the coordinate system.

### 3.2 Meteor position specification

To incorporate the dependence of the $\langle u'w' \rangle$ and $\langle v'w' \rangle$ uncertainties on the temporal and spatial characteristics of the meteor distribution, we have based said distributions used in the model on real measurements. For both the BP and Mylor sites, we constructed a composite day of 2D histograms of the meteor position distributions at 5 km spatial and hourly time resolution, using measurements from April-July 2018. These 2D histograms were taken to represent probability distributions for the meteor positions.

The sampling from these probability distributions at the beginning of each realisation was done according the following process:

1.  Prescribe a number of meteor detections for the day of measurements and altitude in question (e.g., 1340 per day at 90 km for the BP radar case).

2.  Use rejection sampling to distribute those meteors across the day, according to the relative number of meteors in each hour in the input probability distribution.

3.  Distribute the number of meteors prescribed in each hour of measurements according to the probability distribution for that hour, again using rejection sampling.

4.  Return to 1., and repeat for the number of days prescribed in this realisation (for results presented in this paper, 1 or 10).

The horizontal position coordinates assigned to each meteor in the probability distribution (and subsequently the model) are based on the distances from the receiver site in Transverse Mercator coordinates, calculated using the method of Bowring (1989). The altitudes assigned to the meteors are derived from a uniform probability distribution, with a centre value of 90 km and a full-width of 2 km (such that the simulation emulates the idea of analysing meteors from a single height bin).

### 3.3 Meteor detection rate specification

To clarify the effect of a variable number of meteor radial velocity/AOA pairs on the covariance error distribution, a variety of meteor detection rates have been simulated. We have endeavoured to make the detection rates used resemble the number



**Table 2.** Meteor detection rates used for the simulations in this paper. The rates shown are per day, in 2 km-wide bins centred at the altitude specified.

| Altitude (km) | BP | Mylor |
|---|---|---|
| 76 | 140 | 20 |
| 80 | 510 | 130 |
| 82 | 780 | 180 |
| 84 | 1080 | 380 |
| 86 | 1360 | 540 |
| 88 | 1480 | 640 |
| 90 | 1340 | 690 |
| 92 | 1010 | 640 |
| 96 | 300 | 350 |

of meteors detected across a range of heights by the combined BP-Mylor radar link (we note again though that the simulation itself is performed around a single altitude). The detection rates we have used for different heights, listed in Table 2, correspond to those averaged over April 2018 for the two receive sites, in 2 km-wide bins.

### 3.4 Wind field specification

5 The wind field in the simulation is comprised of tidal components and a superposition of monochromatic GWs whose amplitudes have a vertical wavenumber and frequency dependence. Diurnal and semidiurnal tidal components are assumed, with amplitudes of 25 and 10 ms$^{-1}$ respectively. Random phases from a uniform distribution spanning the interval $[0, 2\pi)$ are added to the phase of the zonal component of the tides at the beginning of each realisation, and the meridional component is set to be in quadrature with the zonal component. The 3D wind velocity associated with the GWs at a given time $t$ and Cartesian 10 position vector $\mathbf{r}$ can be written as:

$$\mathbf{v} = \sum_{i=1}^{n_m} \sum_{j=1}^{n_\omega} A(m_i, \omega_j) \mathbf{v}'_{ij} \sin\left(\boldsymbol{\kappa}_i \cdot \mathbf{r} - \omega_j t + \phi_{ij}\right), \tag{1}$$

where $m$ is the vertical wavenumber, $\omega$ is the wave's angular frequency, $n_m$ and $n_\omega$ are the number of vertical wavenumbers and angular frequencies respectively in the spectral grid, $A$ is the joint vertical wavenumber-angular frequency spectral amplitude, $\mathbf{v}' = [u', v', w']$ is the vector of wind component fluctuation sizes, $\boldsymbol{\kappa} = [k, l, m]$ is the 3D wave vector, and $\phi$ represents a 15 (random for each unique $[m_i, \omega_j]$ pair) phase offset.

As per Sect. 3.2, the coordinate system used to specify horizontal position with respect to a reference location (i.e., that embodied by the $\mathbf{r}$ vector) is based on the Transverse Mercator distances evaluated using the Bowring (1989) method (which follow the Earth's surface and take into account its ellipsoidal shape). This is used in preference to line-of-sight distances, the use of which would result in "stretching" of the horizontal scales of the waves at large distances from the coordinate system





origin. Furthermore, the calculated wind velocities are assumed to be in the local East-North-Up (ENU) coordinates at the associated meteor positions.

To ensure that the correlations between the horizontal and vertical winds take on physically reasonable values, we have allowed the component fluctuation amplitudes to be related by the well-known linear GW polarization relations, i.e. $w' = \frac{v_h k_h}{m}$,

where $v_h = \sqrt{u'^2 + v'^2}$ and $k_h = \sqrt{k^2 + l^2}$. The horizontal components are determined by the wave propagation azimuth $\varphi$, through the relations $[k, l] = k_h[\sin\varphi, \cos\varphi]$ and $[u', v'] = v_h[\sin\varphi, \cos\varphi]$.

In order to give the wind field a level of "spatially-correlated randomness" akin to what is seen in mesospheric wind fields when no predominant wave scales are present, we have opted to let $A(m, \omega)$ take on values from a gravity wave spectral model. The vertical wavenumber spectrum we have used (Gardner et al. (1993), eqn. (7), and following their nomenclature) is given

by:

$$F_u(m) = 2\pi\alpha N^2 \begin{cases} m_*^{-3}(\frac{m}{m_*})^s & m \leq m_* \\ m^{-3} & m_* \leq m \leq m_b \\ m_b^{-3}(\frac{m_b}{m})^{5/3} & m_b \leq m \end{cases}, \tag{2}$$

where $m$ is the vertical wavenumber of the wave, and following Gardner et al. (1993), Fig. 1, we let $\alpha = 0.62$, $N = \frac{2\pi}{3\times10^2}\,\mathrm{s}^{-1}$, $m_* = \frac{2\pi}{1.5\times10^4}\,\mathrm{m}^{-1}$, $m_b = \frac{2\pi}{5\times10^2}\,\mathrm{m}^{-1}$, and $s = 2$. The frequency spectrum we have used (Gardner et al. (1993), eqn. (24)) is given by:

$$B(\omega) = \frac{p-1}{f}\left(\frac{f}{\omega}\right)^p, \tag{3}$$

where $\omega$ is the angular frequency of the wave, and following Gardner et al. (1993), Fig. 2, we let $f = \frac{2\pi}{7.2\times10^4}\,\mathrm{s}^{-1}$ and $p = 2$. We then simply assume that the joint vertical wavenumber-angular frequency spectrum is given by the product of these two spectra, i.e.,

$$A(m, \omega) = F_u(m)\,B(\omega). \tag{4}$$

The 2D spectrum we used for results presented in this paper consisted of 80 different vertical wavelengths and wave periods, spanning the ranges 0.5–20 km and 5–240 minutes (uniformly sampled in vertical wavenumber and frequency), respectively. These limits largely encompass the waves responsible for the majority of the momentum deposition in the MLT-region (see e.g., Fritts and Alexander (2003)), whose momentum fluxes are of principal interest in this study.

The wave propagation azimuths were sampled from a uniform random distribution spanning $[0, 180°]$ in bearing, with the

intention being to emulate a wave field whose westward-propagating waves have been removed from the spectrum through selective filtering. This led to true values for the estimates of $\langle u'w' \rangle$ that were on average positive, and values of $\langle v'w' \rangle$ that were on average zero. Testing a wider variety of wave field configurations was considered beyond the scope of the paper.



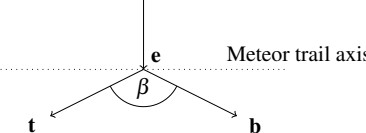

**Figure 3.** Bistatic meteor geometry. Following the nomenclature in Protat and Zawadzki (1999), $\mathbf{t}$ is a unit vector in the direction of the meteor to the transmitter, $\mathbf{b}$ is a unit vector in the direction of the meteor to the bistatic receiver, and $\mathbf{e}$ is a unit vector that is perpendicular to the meteor trail axis and a bisector of $\mathbf{t}$ and $\mathbf{b}$. $\beta$ is the so-called "forward scatter" angle.

The absolute values taken by $A(m, \omega)$ were normalized in a way that resulted in mean values of $\langle u'w' \rangle$ in the vicinity of $20 \ \mathrm{m}^2\mathrm{s}^{-2}$, which is a typical value for this parameter in the MLT-region (see e.g., the discussion in Fritts et al. (2012a)). An example distribution of "true" covariances evaluated in the simulation is shown in the lower panels of Fig. 5.

### 3.5 Projection of the wind velocity onto the Bragg vector

A diagram summarising the bistatic reception geometry is shown in Fig. 3. Following the development of Protat and Zawadzki (1999), the so-called "radial" velocity measured by a bistatic receiver corresponds to the projection of the 3D wind velocity onto $\mathbf{e}$ (also referred to as the Bragg vector in e.g., Stober and Chau (2015)), in turn projected onto $\mathbf{b}$. Mathematically, this velocity is expressed as:

$$v_{\mathrm{r}} = \cos(\beta/2) . \ \mathbf{v}_{\mathrm{ecef}} \cdot \mathbf{e}, \tag{5}$$

which is the velocity that is used to produce a phase progression in the simulated receiver time series, discussed in Sect. 3.6. It should be noted that $\mathbf{t}$, $\mathbf{b}$ and $\mathbf{e}$ are expressed in Earth-centred, Earth-fixed (ECEF) coordinates, and that the wind velocities $\mathbf{v}$ computed in the simulation are in the local ENU coordinates of each meteor. The "ecef" subscript on $\mathbf{v}$ is to denote $\mathbf{v}$'s rotation to the ECEF coordinate system; we have followed the approach discussed in detail by Stober et al. (2018) to do this.

### 3.6 Receiver time series generation and parameter re-estimation

To ensure that realistic radial velocity and position estimation errors are propagated to the covariance estimation, we have opted to generate synthetic receiver time series based on the observables discussed in the previous sections, and to then attempt to re-estimate the observables from the time series. The complex time series for the $j$th receiver is written as:

$$V_j(t) = e^{i(2\pi \mathbf{A} \cdot \mathbf{d} - 4\pi v_{\mathrm{r}} t/\lambda + \Phi_j)} e^{-t/\tau} + n_j(t), \tag{6}$$

where $\mathbf{A} = [\sin\theta\sin\phi, \sin\theta\cos\phi, \cos\theta]$ (where $\theta$ and $\phi$ are the zenith and azimuth angles of the meteor, respectively, as measured from the receiver), $\mathbf{d}$ is a three-element vector of Cartesian displacements to the receiver antenna in question, $\lambda$ is





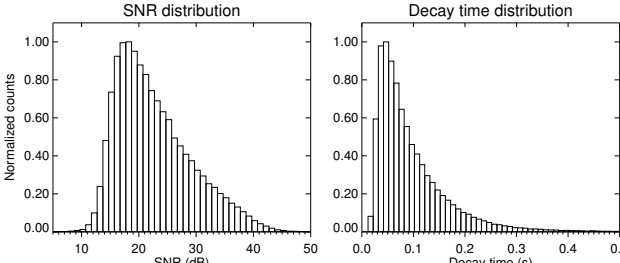

**Figure 4.** Probability distributions of SNR and decay time used in producing the receiver time series discussed in Sect. 3.6.

the radar wavelength, $\Phi_j$ is a phase calibration offset for the $j$th receiver, $\tau$ is the $e^{-1}$ decay time of the meteor, and $n_j(t)$ is a background noise function.

The background noise function consists of values derived from a Gaussian distribution, with a root-mean-square (RMS) value derived from a probability distribution of meteor echo SNRs from the monostatic 55 MHz meteor radar at BP. The values

used for $\tau$ are also derived from a probability distribution from this radar's data. In both cases, the data used to generate the probability distributions spanned 1-30 April 2018, and altitudes 70-110 km. Plots of these distributions are shown in Fig. 4.

The phase calibration offsets $\Phi_j$, which are set for each receiver at the beginning of each simulation realisation, are intended to embody the consequences of incorrectly estimating the true phase calibration offsets between the receiver channels. Based on the phase calibration offset time series shown in Fig. 2, we have chosen to apply to each receiver Gaussian-distributed phase

offsets with an RMS value of $2°$.

Radial velocities and meteor positions are estimated from the noise and phase-offset time series following the procedures outlined in Holdsworth et al. (2004) (Sects. 3.11 and 3.12, respectively), with the exception that the radial velocity is corrected for the forward scatter angle in the case of bistatic reception. Using the definitions in Fig. 3 and following the approaches outlined in Stober et al. (2018) to compute the $\mathbf{t}$ and $\mathbf{b}$ (and $\mathbf{e} = \frac{\mathbf{t+b}}{|\mathbf{t+b}|}$) vectors, the forward scatter angle may be estimated

using:

$$\beta = \cos^{-1}\left(\frac{-\mathbf{t}\cdot\mathbf{b}}{|\mathbf{t}||\mathbf{b}|}\right), \tag{7}$$

and then eqn. (5) may be rearranged for $\mathbf{v}_{\text{ecef}}\cdot\mathbf{e}$, to get said radial velocity.

It should be noted that in rare cases, it becomes impossible to estimate the AOA of the meteor unambiguously when the phase biases and noise are incorporated into the receiver time series (i.e., the error code 3 discussed in Holdsworth et al. (2004)

is encountered). In these particular cases, the echo in question is simply discarded from the subsequent calculation of mean winds and covariances.





### 3.7 Mean horizontal wind and tidal component estimation

The way we have estimated mean horizontal winds in this simulation is similar to that typically applied to meteor radars in the literature (e.g., Holdsworth et al. (2004)). Our approach has been to use singular value decomposition (SVD) to solve the following inverse equation in the least squares sense for $\mathbf{v}$:

$$\mathbf{v_r} = \mathbf{A}\mathbf{v}, \tag{8}$$

where $\mathbf{v_r}$ is a $n_{met} \times 1$ vector of radial velocities ($n_{met}$ being the number of meteors in the time bin under consideration), $\mathbf{v}$ is a $2 \times 1$ vector of wind velocities, and $\mathbf{A}$ is a $n_{met} \times 2$ matrix whose rows take the same form as that described in eqn. (6) (without the vertical component). However, it is important to note in this case that the $\theta$ and $\phi$ defined in $\mathbf{A}$ need to be modified to account for the small rotation of the ENU coordinate systems between the receiver and meteor positions (and be valid for
the meteor's position). The rotation that is to be applied to $\theta$ and $\phi$ is outlined by Stober et al. (2018), App. A4.

In order to remove outliers from the input radial velocity distribution, we follow the iterative scheme proposed by Holdsworth et al. (2004). This involves performing an initial fit for the wind velocities, removing the radial velocities whose value differs from the horizontally-projected radial wind by more than 25 ms$^{-1}$, and repeating the procedure until no outliers are found or until less than 6 meteors remain.

### 3.8 Removal of background wind and tides

To remove the previously estimated mean winds and tides from the time series, we have calculated a low-pass filtered version of the hourly-averaged wind time series using an inverse wavelet transform with a Morlet wavelet basis, and have subtracted the projection of this on the radial velocity time series. This is in principle similar to the approach of Fritts et al. (2010a), who applied an S-transform (in preference to a least squares sinusoidal fit) in order to more completely remove transient spectral
features around the tidal periods from the time series. The application of the inverse wavelet transform is described in App. A.

To ensure that the filtered time series pertain to tidal (or longer)-like wind oscillations (and not short-period GWs), we select a minimum scale size in the reconstruction of 6 hours and a total number of scales of 250. The reconstructed time series is then interpolated to the times of each of the meteors in question, and the radial component of this wind at each of the meteor positions is subtracted from the measured radial velocity.

### 3.9 Covariance estimation

Following the removal of the mean and tidal components of the horizontal wind from the radial velocities, covariances that pertain predominantly to gravity wave-driven wind perturbations are estimated. The approach we apply is identical to that





presented by Thorsen et al. (1997) and subsequently Hocking (2005); much like in the wind estimation, it involves using SVD to least-squares solve the following inverse equation:

$$\mathbf{v}_{\mathrm{r}}'^2 = \mathbf{A}'\mathbf{v}', \tag{9}$$

where $\mathbf{v}_{\mathrm{r}}'^2$ is a $n_{met} \times 1$ vector containing the squares of the perturbation component of the radial velocities,

$$\mathbf{v}' = \left[\langle u'^2 \rangle, \langle v'^2 \rangle, \langle w'^2 \rangle, \langle u'v' \rangle, \langle u'w' \rangle, \langle v'w' \rangle\right]^{\mathrm{T}}$$

is the vector of covariance components, and $\mathbf{A}'$ is a $n_{met} \times 6$ matrix whose rows read:

$$\sin^2\theta\sin^2\phi, \sin^2\theta\cos^2\phi, \cos^2\theta, \sin^2\theta\sin 2\phi,$$
$$\sin 2\theta\sin\phi, \sin 2\theta\cos\phi].$$

It is noted that, as per the wind estimation case, the $\theta$ and $\phi$ terms must be defined according to the orientation of the ENU coordinate system at the meteor in question, not the receive site.

A two-step radial velocity outlier rejection procedure is utilized to remove dubious square radial velocity/AOA pairs from the input distribution in an attempt to reduce the bias in the resulting covariance estimates. The first step is to discard all radial velocity/AOA pairs that have a projected horizontal velocity of $\geq 200$ ms$^{-1}$ (by virtue of which we argue that measured horizontal velocities above this threshold are "nonphysical"). The second step iteratively discards the pairs that satisfy the criterion:

$$|v_{\mathrm{r}i}'^2 - v_{\mathrm{rp}i}'^2| \geq \left[\mathrm{median}\left(\sqrt{|\mathbf{v}_{\mathrm{r}}'^2 - \mathbf{v}_{\mathrm{rp}}'^2|}\right) + \right.$$
$$\left. 5 \times 1.4826 \times \mathrm{MAD}\left(\sqrt{|\mathbf{v}_{\mathrm{r}}'^2 - \mathbf{v}_{\mathrm{rp}}'^2|}\right)\right]^2 \tag{10}$$

where $v_{\mathrm{rp}i}'^2 = A_i'*\mathbf{v}'$ is the $i$th "projected" square radial velocity, "MAD" indicates the median absolute deviation operator, and 1.4826 is the factor to convert a MAD to a standard deviation, assuming the input has a Gaussian distribution. In practice, we have found that the 5 "standard deviations" criterion removes outliers that are large enough to substantially bias the resulting covariance estimates, without iteratively removing an excessive number of samples that are "good". The intention of using the median and MAD statistics (as opposed to mean and standard deviation) has been to reduce the bias outlying points inflict on the "measured standard deviation" of the distribution of $|v_{\mathrm{r}i}'^2 - v_{\mathrm{rp}i}'^2|$.

The performance of the second outlier rejection criterion on simulated data is briefly summarised in Sect. 3.11.3.

### 3.10 "Truth value" of the simulated covariances

To evaluate the "truth value" of the simulated covariances—i.e., that used to estimate the accuracy and precision of the covariances derived through inversion of eqn. 9—we have opted to compute the covariances at the origin of the coordinate system





at 2 min time resolution. This estimate represents what one would measure with an "anemometer" at some fixed location in the vicinity of the radar. We found this estimate to agree extremely closely with that computed at the positions and times of the meteors incorporated in the simulation, which in turn represents the most accurate and precise estimate one could hope to obtain when inverting eqn. (9).

In the case of using wave fields generated from the previously discussed gravity wave spectral model, we found that the covariances estimated by inverting eqn. (9) are more correlated with those calculated using the above two methods than those computed by summing the covariances associated with each wave in the spectrum. Therefore, while the latter method gives the covariances that would be measured over an infinitely large sampling area/time (in a sense the "expectation value" of the covariances), we have refrained from using it as a "truth" value with a view to not overestimating the size of the simulated

technique's measurement errors.

## 3.11    Simulation results

### 3.11.1    Spectrum of gravity waves

This section considers the covariance bias distributions associated with a wind field generated using the GW spectral model discussed in Sect. 3.4. Three different time integration cases (that are later employed in this paper on real data) are tested: 1 day

(which could be considered fairly "high time resolution" sampling of day-to-day variations), 10 days (which sacrifices time resolution for measurement precision), and a 20-day composite (which intends to gather enough meteors in each time-of-day bin for a precise covariance estimate, but in doing so ignores day-to-day variations entirely).

**1-day integration**

The biases for 15,000 realisations of 1-day integrated covariance estimations are shown in Fig. 5. It is clear that the $\langle u'w' \rangle$

term is systematically underestimated, with larger biases present at lower count rates. The width of the bias distribution is also larger at lower count rates. For a simulated mean $\langle u'w' \rangle$ value of $\sim 21$ m$^2$s$^{-2}$, the distribution widths imply a 1-sigma measurement uncertainty of $\sim 65\%$ at the peak of the height distribution, and $\sim 145\%$ at the edges of the distribution, for a multistatic configuration. The same uncertainties are $\sim 72\%$ and 168%, respectively, for a monostatic configuration.

    The width of the bias distributions for $\langle v'w' \rangle$ are also essentially identical to those for $\langle u'w' \rangle$. The relative uncertainties in

the measurements of this term are meaningless, as the wave propagation directions have been chosen in a way that the mean truth value of $\langle v'w' \rangle$ is zero. What the results do illustrate, however, is that there is no bias in the case of estimating a covariance with a zero mean, and that there is no change in the measurement uncertainty of the two components arising from the temporal and spatial distribution of the meteors.

    It should be noted that $\langle u'w' \rangle$ is systematically underestimated for both configurations and for all count rate sets investigated,

especially at lower count rates (the absolute error ranges from about 20% to 50%). Subsequent investigation has confirmed that this occurs when an attempt is made to remove the tidal effects incorporated in the simulated wind field (i.e., the tides are largely removed, but so is some of the variance due to the GWs). The larger biases at low count rates arise from the inability





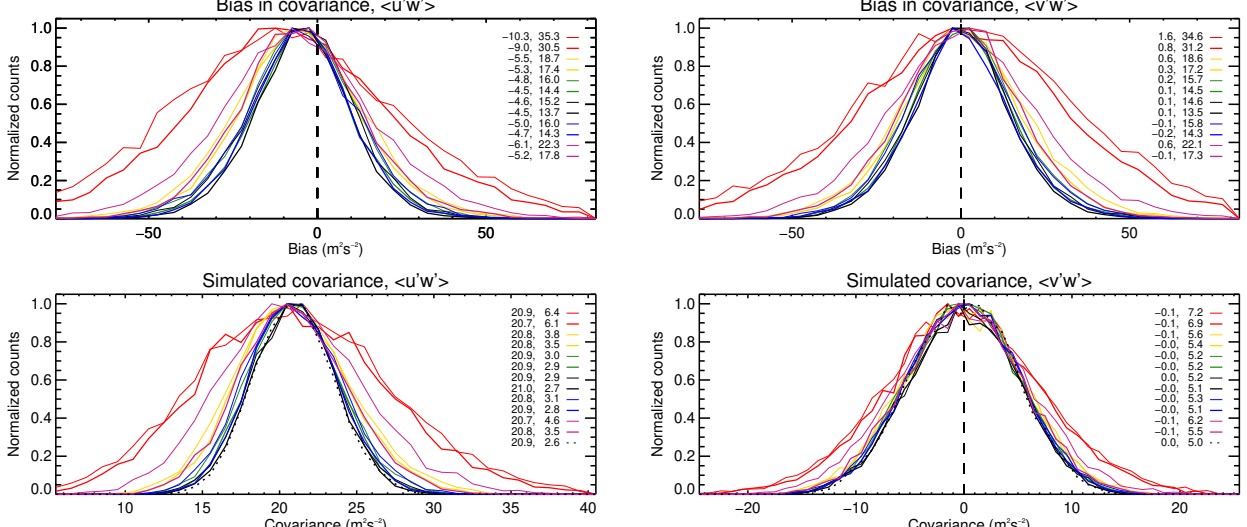

**Figure 5.** Simulated wind covariance bias distributions for 1 day of integration (upper row) and the simulated covariance distributions (lower row). As discussed in Sect. 3.10, biases are calculated with respect to a "reference" value computed at 2 min resolution at the coordinate system origin. The lower row shows the distribution for the "reference" covariance in a dotted black line, and the "true" covariances in coloured lines. The different line colours in each plot represent different simulated heights, which are a subset of those shown Table 2 (red represents 76 km, yellow 80 km, green 84 km, black 88 km, blue 92 km, and violet 96 km). Thick lines show the distribution for the multistatic case (i.e., by combining data from BP and Mylor), and thinner lines show the monostatic case (i.e., just BP data). The mean and standard deviation evaluated from the samples' MAD are shown in the left and right columns respectively of the arrays of numbers in each plot figure.

to define the tidal amplitudes and phases correctly in the presence of wind estimates with larger uncertainties and/or missing wind estimates for particular time bins. Overall, we consider the bias an unavoidable consequence of ensuring that tidal effects are not included in the measured covariances. Further discussion of this point is taken up in Sect. 5.2.

It also appears that there is no clear dependence of covariance uncertainty on the use of a monostatic or multistatic con-

5 figuration, for a fixed detection rate. This is evidenced by the uncertainties at 84 km for the multistatic configuration (1460 detections) being 14.4 $m^2s^{-2}$ and 14.5 $m^2s^{-2}$ for $\langle u'w' \rangle$ and $\langle v'w' \rangle$ respectively, and the corresponding uncertainties at 88 km for the monostatic configuration (1480 detections) being 15.2 $m^2s^{-2}$ and 14.6 $m^2s^{-2}$. In other words, since these uncertainties are essentially the same, we surmise that the multistatic configuration only offers a lower measurement uncertainty at a given height because of the higher number of meteor detections, not because of the altered Bragg vector distribution associated with

10 having two receiver sites.





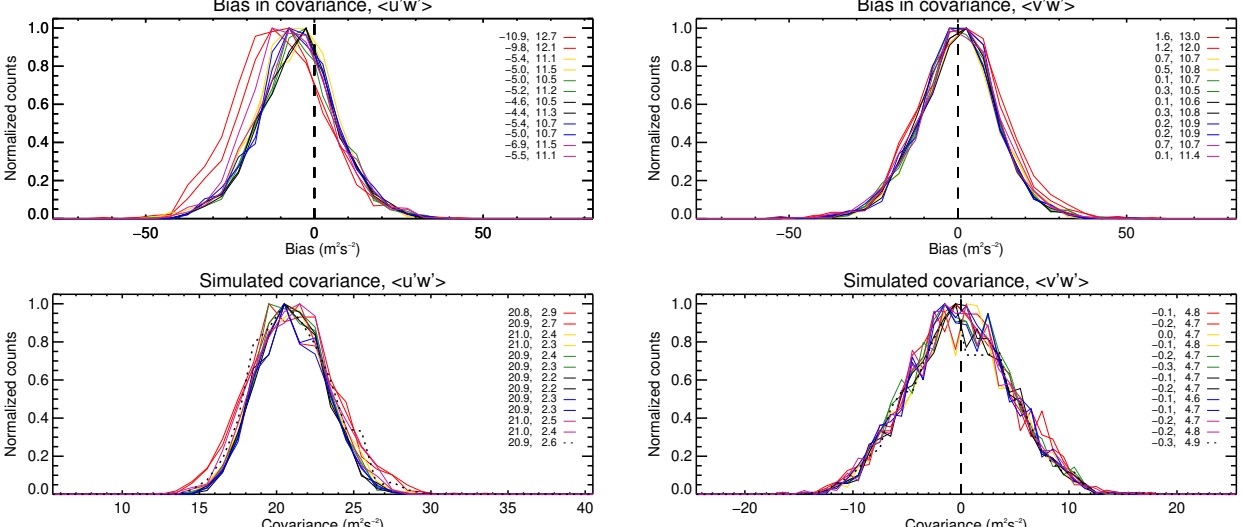

**Figure 6.** As per Fig. 5, but for 10 days of integration.

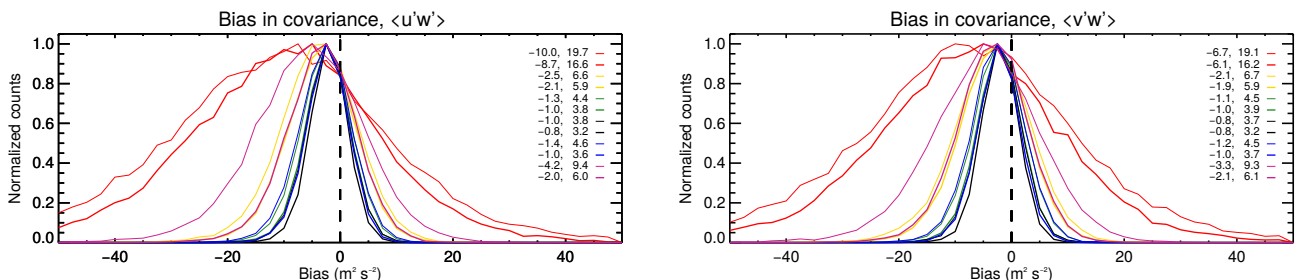

**Figure 7.** As per Fig. 5, but for single monochromatic GWs.

**10-day integration**

Figure 6 shows the bias distribution for 1,500 realisations of 10-day integrated covariance estimates. It is clear that the relative uncertainties in both $\langle u'w' \rangle$ and $\langle v'w' \rangle$ are considerably smaller than for 1 day's integration, ranging from $\sim$50% at the peak of the distribution, to $\sim$60% at the edges. Interestingly, it appears as though the uncertainty is asymptoting to a minimum value, implying that the use of integration times longer than 10 days will lead to diminishing gains in measurement precision. For this reason, we have not opted to use integration times longer than this in the analysis of the BP-Mylor data in this paper.

As per the 1-day integration case, $\langle u'w' \rangle$ has been systematically underestimated, increasingly so at low meteor detection rates. There is also no clear advantage or disadvantage associated with using the bistatic receiver, meteor detection rates aside.



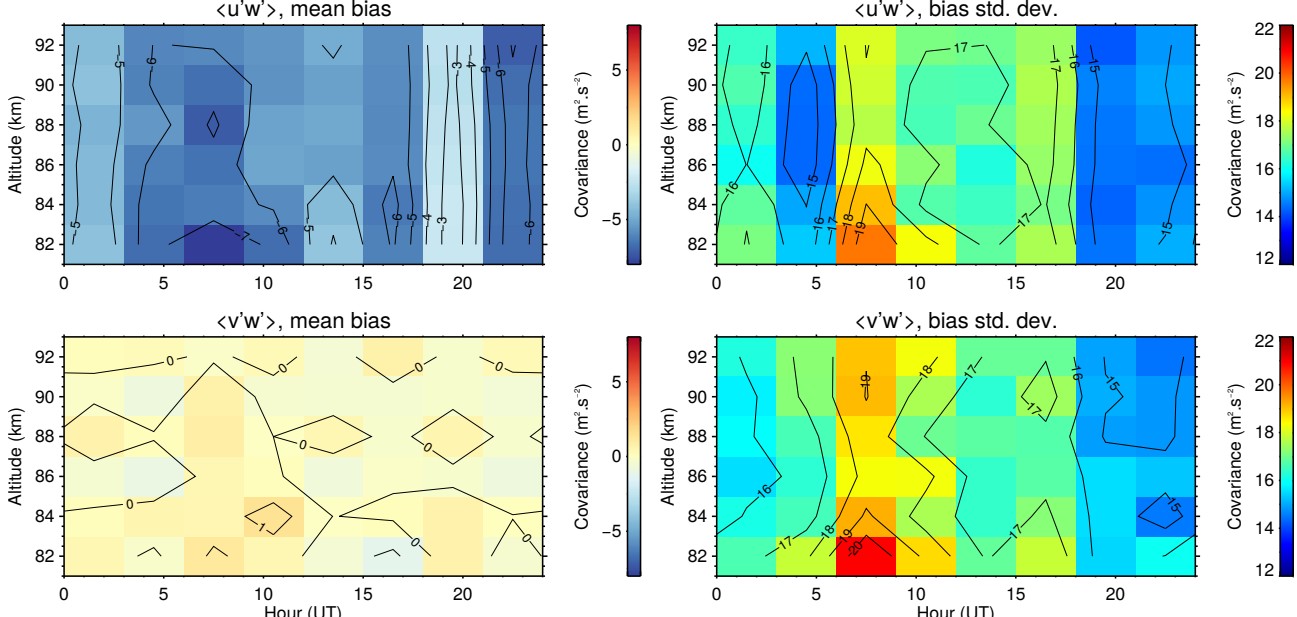

**Figure 8.** Means (top row) and standard deviations (lower row) of the simulated covariance bias distributions for a 20-day composite, as a function of height, for the BP-Mylor link.

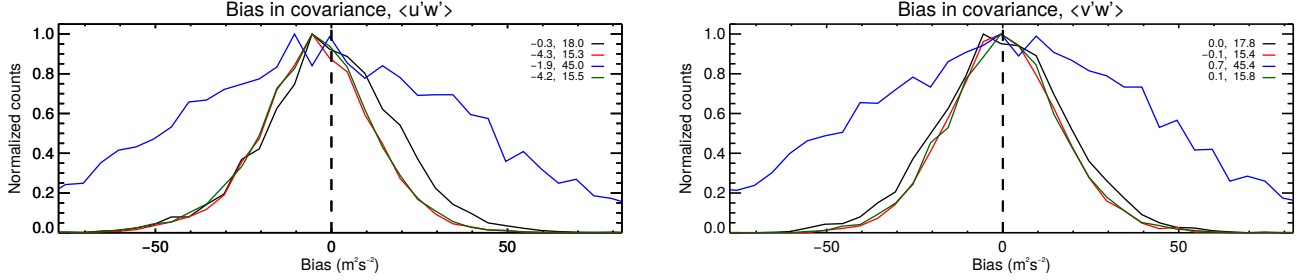

**Figure 9.** Covariance bias distributions for different combinations of outlier contamination and outlier rejection. Black is no rejection or outliers, red is rejection with no outliers, blue is outliers without rejection, and green is outliers with rejection.

## 20-day composite

Figure 8 shows expected values of the covariance bias' mean and standard deviation for 300 realisations of a composite day spanning an interval of 20 days, with three hour time bins, as a function of height from 82-92 km. The highest standard deviations for both $\langle u'w' \rangle$ and $\langle v'w' \rangle$ occur in the 6-9 and 9-12 UT bins, and the lowest in the 18-21 UT bin. The mean value for $\langle u'w' \rangle$, which is again $\sim 21$ m²s⁻², implies a relative uncertainty at the peak of the height distribution of about 70% in the 18-21 UT bin, and about 85% in the 6-9 UT bin. It should be noted that the uncertainty is as high as $\sim 100\%$ in the 6-9 UT bin at 82 km.



Once again, a systematic underestimation of $\langle u'w' \rangle$ is present, which as discussed in is Sect. 3.11.1 is an artefact of attempting to remove tidal effects.

### 3.11.2 Monochromatic gravity wave

The previous section considered a wind field containing a multitude of waves whose spatial/temporal scales spanned a large

part of the spectrum atmospheric gravity waves are expected to occupy. This section briefly addresses the other limiting case, which is that of a wind field consisting of a single monochromatic wave.

In all simulation realisations for this case, we have set the single monochromatic wave's propagation direction to 45°T, so as to make the true $\langle u'w' \rangle$ and $\langle v'w' \rangle$ covariances equal. A horizontal wavelength and phase speed has been randomly selected for each realisation, from a uniform distribution with bounds [10, 60] km and [10, 40] ms$^{-1}$, respectively. A 1-day integration

is used for the covariance estimate.

The bias distributions for 15,000 realisations are shown in Fig. 7. As per the spectral wave field case, the distribution widths are largest at the edges of the height distribution, and narrowest at the peak. However, the widths are far smaller than in the spectral wave field case. Across all wavelengths and phase speeds, the simulated mean true covariance was $\sim$38 m$^2$s$^{-2}$, which translates to uncertainties of about 8% and 44% at the peak and lower edge of the height distribution respectively for the

multistatic configuration. For the monostatic configuration, the same uncertainties are about 10% and 52%, respectively.

Similarly to the spectral wave field case, both covariance terms are systematically underestimated (ranging from about 2% to 26% for $\langle u'w' \rangle$ in the multistatic configuration, at the peak and lower edge of the height distribution, respectively). Interestingly, $\langle v'w' \rangle$ is underestimated to a slightly lesser degree than $\langle u'w' \rangle$. Once again, there is also no clear advantage or disadvantage of using the bistatic receiver (meteor detection rates aside).

### 3.11.3 Outlier rejection criteria performance

This section shows the effect of the application of the outlier rejection criterion of eqn. (10), in the absence of tidal effects and attempted removal of them.

To emulate a radial velocity time series "partially corrupted" with outliers in this section, Gaussian-distributed noise with a standard deviation of 50 ms$^{-1}$ has been added to a randomly selected 5% of the radial velocity estimates in a given realisation.

We note that radial velocity errors of this size are rare in practice; they have been used to test the rejection criterion's robustness, and to allow us to highlight potential downsides of not having the criterion in place.

Figure 9 shows the covariance bias distributions for the same spectral gravity field as applied in Sect. 3.11.1 and for 1 day of integration, for four cases: rejection not applied with no outliers present, rejection applied with no outliers present, rejection not applied with outliers present, and rejection applied with outliers present. The mean true values for $\langle u'w' \rangle$ and $\langle v'w' \rangle$ are

the same as in Sect. 3.11.1, i.e., $\sim$ 21 and 0 m$^2$s$^{-2}$, respectively.

The application of the criterion is clearly beneficial in the presence of outliers, resulting in a reduction in relative uncertainty of the $\langle u'w' \rangle$ estimate from about 214% to 74%. Interestingly, the application of the criterion in the presence of no outliers





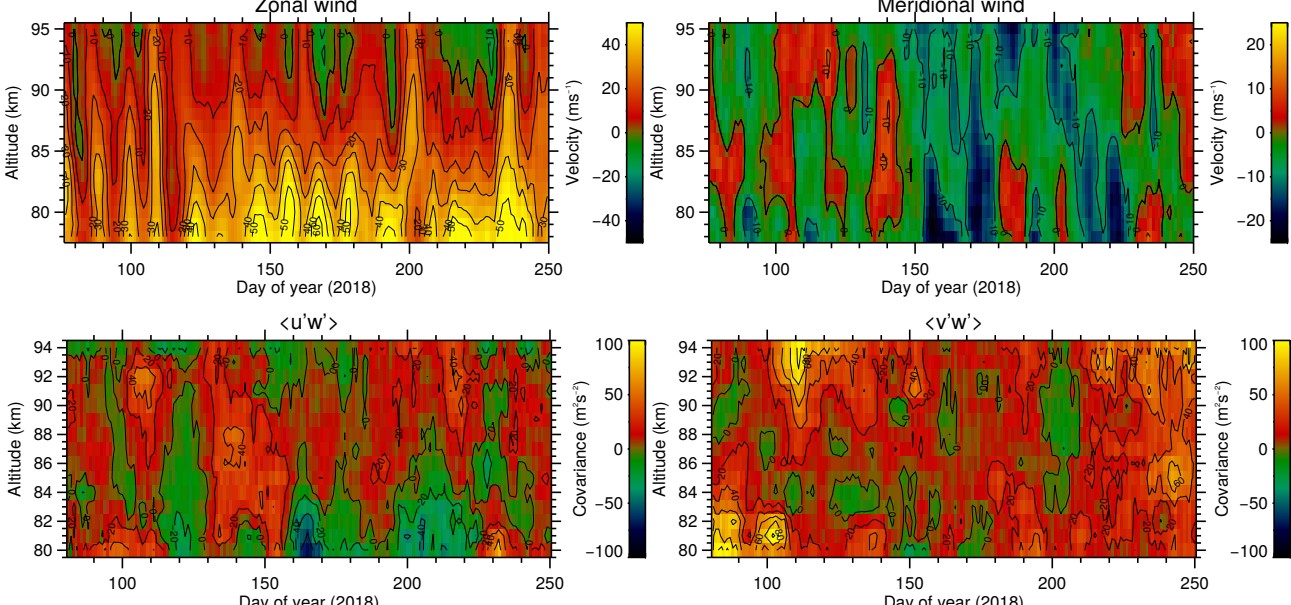

**Figure 10.** Mean horizontal winds (top row) and the $\langle u'w' \rangle$ and $\langle v'w' \rangle$ covariance components measured using the BP-Mylor link between 17th March and 9th September 2018. As discussed in Sect. 4.1, the winds shown correspond to a 10-day moving average of the hourly-averaged winds with tidal components removed, and the covariances have been evaluated over 10-day windows, with a gap of 2 days between the centres of adjacent windows.

also results in a slight reduction in relative uncertainty (from about 86% to 73%), although it does result in $\langle u'w' \rangle$ being underestimated (by about 20%). This point is revisited in Sect. 5.3.

Despite the fact that it appears to introduce a small measurement bias, we still apply the criterion in the subsequent analysis of BP-Mylor data, so that we can be assured that anomalous radial velocities do not contribute to the covariance measurement
5  errors.

## 4  Momentum flux retrievals

This section uses the methodology described in the previous section to estimate covariances from the BP-Mylor meteor radar link from 17th March 2018 through to 9th September 2018. The aim of this analysis was originally to verify that the estimated covariances and flow acceleration derived from them were physically reasonable; however, in observing an apparent tidal
10  modulation of the covariances, we realised that the results themselves may be of more general interest.





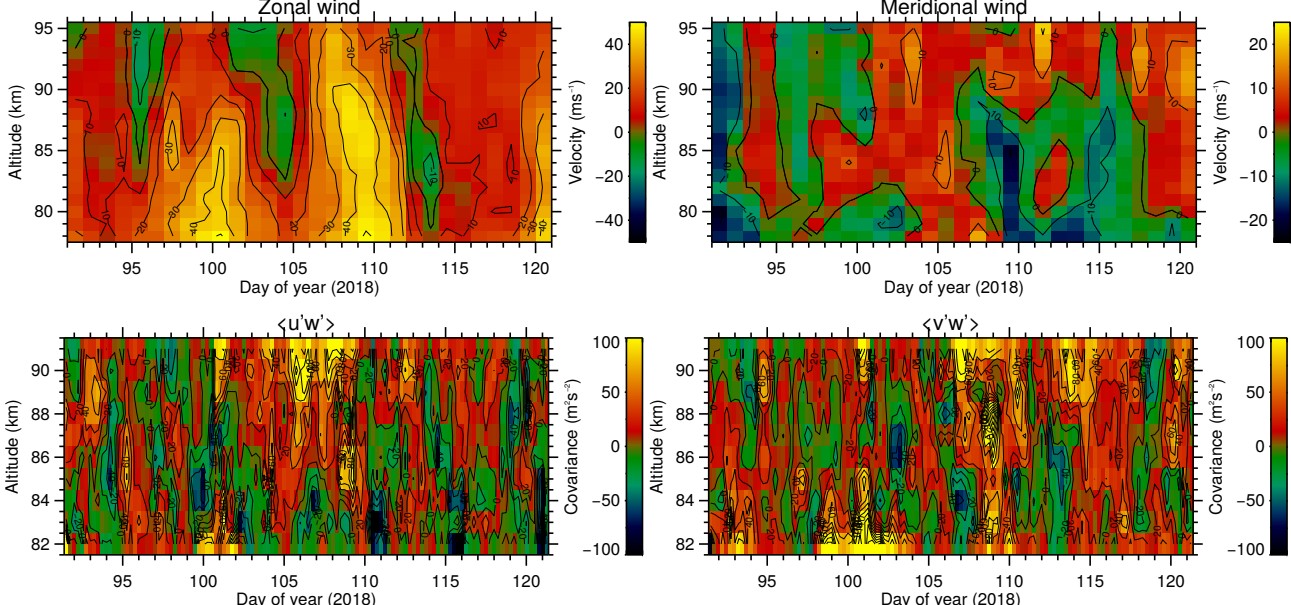

**Figure 11.** As per Fig. 10, but for April 2018. Also, in this case no moving average has been applied on the winds post-tide-removal, and the covariances have been evaluated over windows of length 1 day, with a gap of 6 hours between the centres of adjacent windows.

## 4.1 Covariances during the Austral winter

Plots of the mean horizontal winds and the $\langle u'w'\rangle$ and $\langle v'w'\rangle$ covariance terms from 17th March through to 9th September 2018 are shown in Fig. 10. Both quantities have been sampled using 2 km, non-oversampled altitude bins. We chose to evaluate the covariance terms using 10-day-long windows, with a gap of 2 days between the centres of adjacent windows, in attempt to

5 resolve the planetary-wave induced modulation of the covariances. A low-pass wavelet filter with a cut-off of 2 days and a 10-day moving average has been applied to the hourly horizontal winds to evaluate the winds shown; the filtering was performed to avoid the aliasing of GW activity and tides into the wind's variability, and the moving average in order to more closely match the temporal sampling of the two parameters. Therefore, the winds shown should provide a good measure of the "background mean winds" responsible for selective filtering of the gravity wave spectrum.

As is expected for this time of year at a mid-latitude SH site (see e.g., Vincent and Ball (1981)), the eastward winds around 80 km generally increase with time from the autumnal equinox to the winter solstice ($\sim$ days 80 and 170 respectively) and decrease toward the vernal equinox ($\sim$ day 265). A wavelet analysis (not shown here) reveals that much of the shorter term zonal wind variability evident in the figure is transient, and encompasses a spectrum of periods between about 10 and 60 days. The meridional wind, conversely, has a mean much closer to zero. Much of its variability is confined to periods around 10, 20,

25 and 40-50 days below 90 km, with variability in the 50-100 day period becoming increasingly dominant above 90 km.

The level of (anti)correlation between the covariance terms and the winds is highly variable. The $\langle u'w'\rangle$ term appears to be anticorrelated with the zonal wind between 80 and 84 km around the winter solstice, as does $\langle v'w'\rangle$ with the meridional





**Figure 12.** Amplitude of the diurnal (top row) and semidiurnal (second row) tides, and and phase of the diurnal (third row) and semidiurnal (fourth row) tides as measured by the BP-Mylor meteor radar during April 2018.

wind above 88 km across a similar time interval. While pronounced levels of anticorrelation between these quantities in the mesospheric region arising from the selective filtering mechanism are typical (see e.g., the recent summary provided by Jia et al. (2018))—particularly in the zonal component—departures from these predictions are also not uncommon. As Jia et al. (2018) explains, it is difficult to conceive a mechanism for departures from this theory in the zonal component (given the dominance

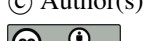



of eastward winds in the lower mesosphere during winter), aside from considering that the GWs may have propagated from a region with weak eastward mesospheric winds.

The feature we focus the remainder of this discussion on concerns the coincident enhancement in the $\langle u'w' \rangle$ and $\langle v'w' \rangle$ terms in the interval spanning days 100 to 120, around 90-94 km. Peak values of $\sim 50$ m$^2$s$^{-2}$ and 100 m$^2$s$^{-2}$ for $\langle u'w' \rangle$ and

$\langle v'w' \rangle$ respectively are obtained during this interval. Interestingly, they coincide with a brief enhancement in the zonal winds at the same height, and the peak of the northward phase of an oscillation in the meridional winds with periods spanning 50-100 days.

Figure 11 shows an inset of Fig. 10, spanning April 2018 (which the aforementioned covariance enhancement is centred on). In an attempt to increase the temporal resolution, the covariances in this figure have been evaluated with 1-day windows, with

a gap of 6 hours between adjacent windows. Tidal components have also been removed from the winds as per Fig. 10 (i.e. in order to not alias tidal/GW activity into the winds), and for a closer match to the time sampling of the covariances, no moving average has been applied.

This figure shows evidence of a pronounced periodicity around 10 days in the zonal wind, which attains its highest amplitude at approximately day 110 around 85 km. At this time and in the same altitude region, the mean meridional winds abruptly (over

a period of a few days) switch from northward to southward. All of this variability is likely attributable to a superposition of planetary waves. Albeit noisy (owing to the relatively short integration time), the $\langle u'w' \rangle$ covariance term shows an enhancement between days 105 and 110, and attains especially high positive values (exceeding 100 m$^2$s$^{-2}$) at around 90 km altitude. Interestingly, the $\langle v'w' \rangle$ enhancement lags that of $\langle u'w' \rangle$ by several days, with a peak again in excess of 100 m$^2$s$^{-2}$ around day 110.

We have also noted that this interval is associated with an abrupt enhancement of the amplitudes of the diurnal and semidiurnal tides. Figure 12 shows the amplitude of the horizontal wind time series reconstructed from a inverse wavelet transform (see eqn. A1), for scales between 0.4 and 0.6 days for the semidiurnal tide, and 0.8 and 1.2 days for the diurnal tide. The diurnal tide in the zonal wind is seen to reach an amplitude of $\sim 50$ ms$^{-1}$ during day 107 at a height of around 92 km, and 35-40 ms$^{-1}$ in the meridional component around 88 km during day 109. It should be noted that the hourly averaged zonal wind

velocity (not shown here) reached a maximum of about 140 ms$^{-1}$ at 92 km during this period. The semidiurnal tide, whose amplitude is known to rarely exceed 10 ms$^{-1}$ at Adelaide's location (e.g., Vincent et al. (1998)), also reached an amplitude of 35-40 ms$^{-1}$ during day 104 in both the zonal and meridional components, at a height of around 94 km. The figure additionally shows that the phase of the diurnal tide is modulated, with the time scale of those modulations appearing to follow the phases of the planetary wave activity in Fig. 11—although there are no noteworthy phase changes at the times of the sudden amplitude

enhancements. The semidiurnal tidal phase is persistent, and also with a well-defined vertical progression, during the few days in which its amplitude is large, but clearly has little meaningful structure at other times.

The large tidal amplitudes during this period lead us to expect the propagation directions of the GWs removed from the wave spectrum by the winds to exhibit a diurnal variation. A complicating factor is that these waves may also amplify, dampen or shift the phase of the tide, depending on the waves retained in the spectrum at the wave breaking height; the large variability

in the tidal amplitudes during this period indicates that this may have indeed occurred. To provide some clarity on the extent





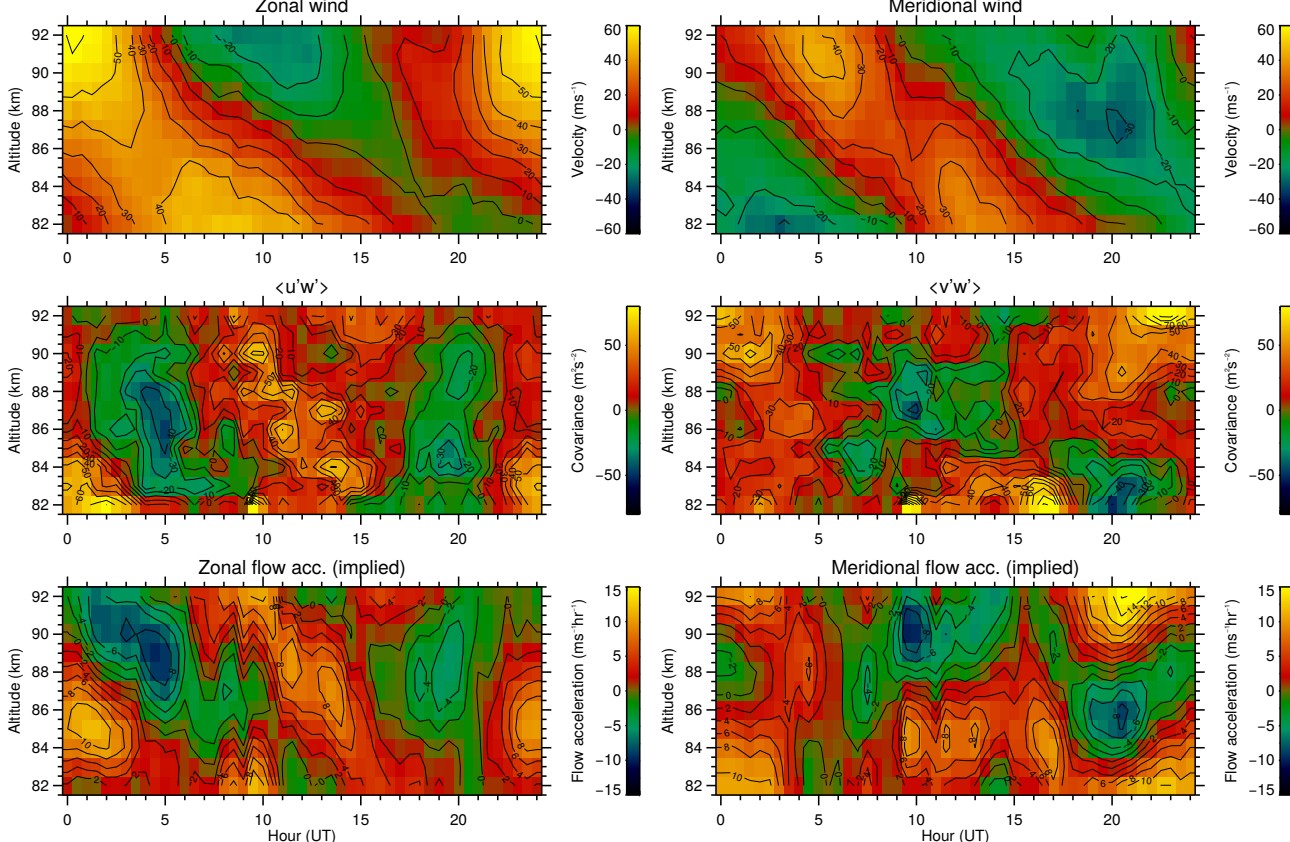

**Figure 13.** A composite day of the horizontal winds (top row), covariances (middle row) and flow accelerations implied by the covariances (lower row), spanning 5-25 April 2018.

to which the GWs have been modulated by the tide and vice versa, in the next section we examine a composite day of the tidal winds, covariances and the implied flow accelerations over a 20-day interval spanning the interval in which the diurnal tide has a reasonably consistent phase and an enhanced amplitude.

## 4.2 Observed GW-tidal interaction

5  Figure 13 shows a composite day of the horizontal winds, covariances, and flow accelerations implied by the covariances, over 5-25 April 2018 (i.e., days 95-115). The composite day consists of time windows of width 3 hours, with a gap of 30 minutes between the centres of adjacent windows. The height binning again consists of 2 km-width bins with centres separated by 1 km.



The flow accelerations (e.g., in the case of the zonal direction) have been evaluated using the expression

$$\langle F_x \rangle = -\frac{1}{\rho(z)}\frac{\partial}{\partial z}\left(\rho(z)\langle u'w' \rangle\right), \tag{11}$$

where $\rho(z)$ represents the neutral density as a function of height $z$. The density climatology we have used has been derived from the Sounding of the Atmosphere using Broadband Emission Radiometry (SABER) satellite instrument (see App. B for details). Similarly to Liu et al. (2013), we also apply a low-pass filter with a cut-off wavelength of 10 km to the vertical profile of the covariance prior to evaluating its density-weighted derivative, in order to remove small-scale fluctuations from it that are clearly not associated with tidal modulation.

As expected from the amplitudes in Fig. 12, both horizontal wind components show a predominantly diurnal variation, with the meridional component lagging the zonal's by approximately 6 hours across the observed height region. The time of the zonal wind maximum occurs around 0 UT at 92 km, and 8-9 UT at 82 km.

In contrast, the $\langle u'w' \rangle$ covariance term shows a predominantly semidiurnal variation with little vertical phase progression, maximising at around 0 and 12 UT, and minimising around 5 and 20 UT. The $\langle v'w' \rangle$ term is more variable with altitude, exhibiting a semidiurnal variation between 82 and 84 km, and a largely diurnal variation above this. The semidiurnal variation between 82 and 84 km is associated with positive covariances for the entire day except between about 18 and 24 UT, and the diurnal variation above is associated with negative values between about 8 and 15 UT and positive otherwise.

Between about 88 and 92 km, the zonal flow acceleration shows a pronounced minimum between 4 and 6 UT, a maximum around 13 UT at about 88 km, and a weaker minimum around 19 UT. The maximum occurs at a similar time to the corresponding zonal wind minimum, whereas the first minimum lags the zonal wind maximum by about 5 hours, and the second minimum precedes it by about 5 hours. Conversely, there is little flow acceleration structure below 87 km, other than a broad maximum at about 85 km around 1 UT. These observations are difficult to reconcile for three reasons: (1) the wave forcing is consistent with a rapid deceleration of the zonal wind from 4-6 UT at around 90 km, but there appears to be no positive forcing around 20 UT to accelerate the wind, (2) the strong positive forcing which does occur around 13 UT appears to result in little wind variability, and (3) the positive forcing around 85 km between 23 and 4 UT is associated with an acceleration of the zonal wind, but this acceleration is much smaller than that around 90 km.

From 88-92 km, the meridional flow acceleration shows a small maximum around 4 UT, a minimum at about 10 UT, and a large maximum around 20 UT. As per the zonal case, this leads to a peculiar relationship with the meridional wind; the forcing's large maximum occurs at a similar time to the wind minimum, the minimum corresponds roughly with a rapid wind deceleration, and the smaller maximum corresponds with a rapid wind acceleration. Also like in the zonal component, there is little meridional flow acceleration structure below around 86 km.



## 5  Discussion

### 5.1  Uncertainties in $\langle u'w' \rangle$ and $\langle v'w' \rangle$ estimates

In the simulations section of this paper, we have tried to conclusively define estimates for the absolute and relative uncertainties of the $\langle u'w' \rangle$ and $\langle v'w' \rangle$ covariance terms as measured by the multistatic BP-Mylor meteor radar, for typical time and height sampling cases. We subsequently replicated these sampling schemes on the case study data. Even with this replication, we have noticed that there are three main caveats in applying the uncertainties directly to the observations:

1. As shown by Kudeki and Franke (1998), the covariance estimation uncertainty is proportional to the geometric mean of the horizontal and vertical variances, in the case of sampling the wind field using a perfect "anemometer". Assuming this holds for a meteor radar-like detection distribution, this means that the absolute uncertainties of $\langle u'w' \rangle$ and $\langle v'w' \rangle$ reported in this paper should be similar for a given wave field, regardless of the value of $|\langle u'w' \rangle|/|\langle v'w' \rangle|$. Therefore, the likelihood of correctly estimating the sign of one component at any given time/height may be very different to the other.

2. As evidenced by the differences in the distribution widths of Figs. 5 and 7 for given detection rates, the relative uncertainties of a non-zero covariance term appear to be dependent on the total frequency/scale span of all the associated waves. In our example, the relative uncertainty in the covariance for a spectral GW field is around 8 times that for a single monochromatic GW. This finding, which is qualitatively consistent with the conclusion reached by Vincent et al. (2010) (for high meteor detection rates) and Fritts et al. (2012a), makes it impossible to accurately define the covariance measurement uncertainty for this radar without a-priori knowledge of the GW field and its variation with time.

3. The spectral components of the wave field may vary during the integration period. This is particularly problematic for the 10-day window; for example, during a period of intense but short-lived monochromatic wave events followed by more "complex" wave activity, increasing the integration time may actually increase the uncertainty in the covariance estimate of the monochromatic wave activity—not only because of the likely change in the mean covariance, but also because of the "noise" added to the radial velocity time series by the more "complex" activity.

Despite these caveats, we can broadly conclude that the 10-day integrated covariances (Fig. 10), except where the absolute values are smaller than about 10-15 $\text{m}^2\text{s}^{-2}$, are likely to be of the correct sign. The correlation length of the features in both the time and height domains also indicates that the noise component in the signal is considerably smaller than the sum of all the modes of geophysical variability. Additionally, at this time integration there is likely to be little difference in the uncertainty at the peak and edges of the height region analysed.

The 1-day integrated covariances (Fig. 11), in contrast, are far more affected by measurement noise. There is still some degree of temporal-height correlation, especially in the region of consistently high values of $\langle u'w' \rangle$ between days 105 and 110 above about 86 km, but very little below 84 km. The excursions below 84 km are of the same order as the simulations predict





for 1 day of integration in a spectral wave field, so it may be that the noise component at these heights is considerably larger than the signal.

The 20-day composite covariances (Fig. 8), while clearly affected by measurement noise, do not show fluctuations from bin-to-bin of the same size as the uncertainties predicted in the corresponding simulation. This gives weight to the covariance

structures observed, and also suggests that the wave field being observed over the 20-day period was not as complex as the simulation's, nor particularly variable.

Unfortunately, it is impossible to know (using the meteor observations alone) if the discrepancies between the 1-day and the 10-day integration (for example, the absolute values of the covariances during the enhancement between days 105 and 110) are a result of statistical noise in the 1-day estimate or a precise estimate of a strong, transient monochromatic wave event using

the 1-day integration. The observation of waves in the MLT airglow may aid in the interpretation of how "monochromatic" the background wave field is; in future, we intend to complement these meteor radar case studies with images of the sodium and hydroxyl airglow taken nearby the BP site. This, in conjunction with the random resampling method employed by Liu et al. (2013), may lead to more refined uncertainty estimates.

In the 10-day integrated results, the small difference in measurement error at the peak and lower edge of the height distribu-

tion (around 20%, for an order of magnitude increase in detections) places an important question on the usefulness of further increasing the integration times/detection rates. On this point, Fritts et al. (2012a) argued that the covariance measurement error should decrease with the square root of the number of detections, and by extrapolating from the 250% error for a 1-hour integration presented in Vincent et al. (2010), concluded that their relative error for a one-month composite should have been as low as 10%. Our simulations suggest that an increase in precision of this magnitude cannot occur. Moreover, using a similar

detection rate and a 3-hour bin in our 20 day composite of a spectral model-derived wave field shown in Fig. 8, we obtain a minimum relative error of about 70%. In saying this, we note of course that a relative error of 10% is possible for a considerably less complex wave field.

## 5.2   Effects of tides on covariance estimates

All of our simulations have shown that a systematic underestimation of non-zero covariances arises when an attempt is made

to remove tidal effects. This clearly becomes more of a problem in the presence of large amplitude GWs with ground-based periods close to those of the tides. A number of questions about the process of tidal removal could be raised:

1. What is the importance of incorporating the momentum fluxes of gravity waves with ground-based periods close to the tides in climate models?

2. If those longer-period waves are unimportant, what is an appropriate frequency cut-off for covariance measurements?

3. If those waves are important, what is the optimal way to remove the tides?

With regard to 3., it may be that a wavelet/S-transform has insufficient frequency resolution to define solely tidal features; a long-windowed harmonic fitting (as used by e.g., Andrioli et al. (2013a)) may be more appropriate if there is a specific interest





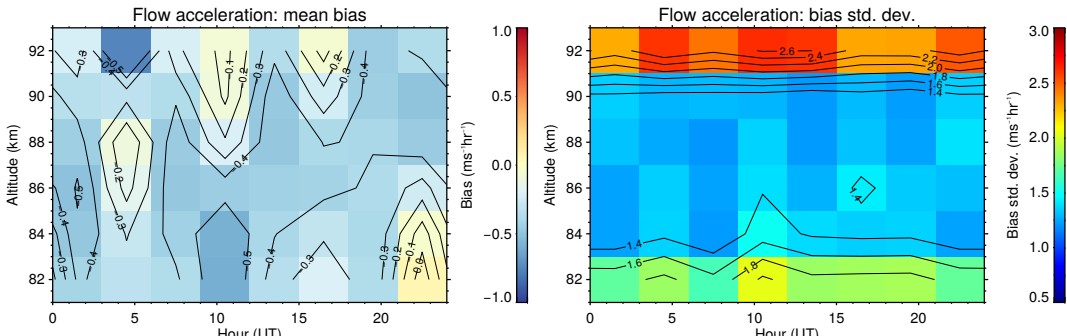

**Figure 14.** Simulated errors in flow acceleration estimates, using the bias mean and standard deviations in the Fig. 8 covariances.

in GW features close to or between the tidal periods. Of course, this method assumes no variability in the tidal amplitudes, tidal periods, or in the GW spectrum. The best way forward may be to simply apply both of the methods independently and contrast their effects.

### 5.3 Radial velocity outlier removal

In Sect. 3.11.3 we showed that the radial velocity outlier rejection scheme of eqn. (10) substantially increases the covariance measurement precision in the presence of outliers. However, we note that the criterion used (especially the "5 standard deviations" aspect) has not been rigorously tested; we merely selected it on the basis of it removing points in the distribution of $|v'^2_{\mathrm{r}i} - v'^2_{\mathrm{rp}i}|$ (real and simulated) that we had noticed were spuriously affecting the covariance estimates. A more rigorous scheme would adaptively modify the thresholding based on observed characteristics of the wind field, rather than simply the
residual of the fit.

A complication arises from the fact that the criterion results in a more precise (albeit less accurate) covariance estimate in the absence of outliers. This also illustrates an important point about the sensitivity of the eqn. (9) inversion to the input: it is as though the data that contribute to the accuracy of the measurement actually increase the measurement's uncertainty, if they are associated with large radial velocity perturbations.

### 5.4 Observed GW-tidal interaction

Our aim in analysing the GW-induced flow accelerations in Sect. 4.2 has been to verify that the estimated momentum fluxes were physically reasonable and devoid of tide-induced biases, and to contribute to the well-known gap in knowledge of GW effects on tides. Our analysis, which was centred on a 20-day interval containing an abrupt enhancement in tidal amplitudes, has yielded inconclusive results on whether the GW momentum deposition has on the whole enhanced, dampened or changed
the phase of the tidal motions. Nevertheless, the expected uncertainties in the flow accelerations based on the bias mean and standard deviations in the Fig. 8 covariances, shown in Fig. 14, indicate that the signal components between 84 and 90 km shown in Fig. 13 will have well exceeded the noise levels.



The results are complex, illustrating tidal enhancement at some times of day, dampening at others, and that there are also times in which a forcing is present but no apparent effect on the tide is clear. A broad observation is that the forcing components have a more pronounced diurnal variability between about 86 and 92 km, with the result that the forcing dampens the tide at the tide's minimum (i.e. westward and southward phase), and shifts its phase at its maximum. Of course, our interpretation is complicated by the fact that we have no knowledge of what the tidal features may have looked like without any GW forcing.

It is widely accepted in modelling studies that GW forcing plays a role in the observed seasonal variation of the migrating diurnal tide (DW1) amplitudes (i.e. equinoctial maxima and solsticial minima), and that whether amplification or dampening of the amplitude occurs depends on the GW source spectrum (e.g., Ortland and Alexander (2006); Yiğit and Medvedev (2017)). However, there is still ongoing debate about whether or not the forcing is responsible for all of DW1's observed amplitude and phase variability. For example, both Mayr et al. (1998) and Watanabe and Miyahara (2009) have concluded that the forcing is in phase with DW1 during the equinoxes and out of phase during the solstices, leading to DW1's amplification at the equinoxes and dampening at the solstices. Yiğit and Medvedev (2017) reached the same conclusion for the September equinox, but stated that Watanabe and Miyahara (2009) may have significantly underestimated the magnitude of the forcing. In contrast, for the March equinox Lu et al. (2012) has argued that the tidal variability is caused by a superposition of GW forcing and advection terms that varies with altitude and latitude, and that GW forcing exclusively dampens tidal amplitudes in the MLT/I. Moreover, Lu et al. (2012) has reported considerably larger GW forcing magnitudes than in a related modelling study by McLandress (2002).

The small number of recent observational studies that have sought to quantify the effect of GW forcing on the DW1 amplitude and phase have also yielded contradictory results. For example, using TIMED satellite data Lieberman et al. (2010) showed that while the zonal and meridional GW forcing maximises at the equinoxes and minimises at the solstices, the zonal forcing is in quadrature with the zonal tidal wind, and the meridional forcing is out of phase with the meridional tidal wind, leading to a zonal tide with advanced phase and a dampened meridional tide. They noted that the zonal advection due to variability in the meridional DW1 amplitude also, like the GW forcing, maximised at the equinoxes and minimised at the solstices, but were not able to reconcile if this variability was a cause or an effect of the seasonal DW1 variation. Also using TIMED data, Xu et al. (2009) concluded that the GW-induced dampening of tidal amplitudes is largest during equinoxes, and therefore that dampening cannot cause the observed seasonal variation in tidal amplitudes. In contrast, using measurements from a ground-based meteor radar in Hawaii (20.7°N, 156.3°W), Liu et al. (2013) noted that GW forcing tends to slightly dampen the DW1 amplitude below 90 km, but enhance it above 90 km. Using a similar approach on LIDAR data from Starfire Optical Range (35.0°N, 106.5°W), Agner and Liu (2015) also noted that GW forcing can amplify or dampen the DW1 amplitudes, depending on the altitude.

Tides may also interact with GWs through the diurnal variations in atmospheric stability they induce (i.e., making conditions more favourable for GW breaking and hence GW forcing at particular times of day). For example, Fritts et al. (1988) showed from observations at Scott Base, Antarctica that the highest levels of turbulence due to convective instability occurred at the times that the vertical component of the tidal wind induced the most negative value of $dT/dz$ (the vertical temperature gradient). Using temperature perturbations from the GSWM-98 model for the BP site, Holdsworth et al. (2001) also showed





that maximum negative values of $dT/dz$ were in phase with the maximum values of the turbulent velocity measured by the BP MF around the autumnal equinox. Using GSWM-00 output, we have noted that the maximum negative $dT/dz$ (of $\sim -1$ K km$^{-1}$) should occur between 1-3 UT across the 85-92 km region at the BP site during the period of our composite day analysis; curiously, we observe large positive values of $\langle F_x \rangle$ at this time just below this region, and an abrupt shift in the sign

of $\langle F_x \rangle$ above it. As Holdsworth et al. (2001) notes, while a $dT/dz$ of this size is too small to result in static instability, it still corresponds with a large level of GW forcing and the maximum eastward phase of the diurnal tide, which we have observed to be particularly large during this interval.

## 6    Conclusions

This study has defined limits on the expected uncertainties in estimates of the $\langle u'w' \rangle$ and $\langle v'w' \rangle$ covariance terms made using a

multistatic meteor radar, and has presented an example case study of using the radar to measure the GW forcing on the diurnal tide that arises from the height variation of the measured covariances. We have concluded that the extra detections offered by the bistatic receiver appreciably improve the precision of the covariance measurements, although little of that improvement can be attributed to the increased Bragg vector diversity associated with having two viewing perspectives. The winds observed in the case study revealed substantial variations in the amplitude of the diurnal tide, but we were unable to conclusively show if

GW forcing caused this variation. Nevertheless, our simulations have indicated that the bulk of the variability in the covariance and GW forcing we have seen far exceeds the expected measurement uncertainties, and therefore that GW forcing has not been the only contributor to the tidal variability. We note that studies concerning GW forcing on tides are few, and that there is a clear need for further studies at other locations. Furthermore, there is a need for a definition of the part of the GW spectrum that is most likely to contribute to forcing on the tides; this will inform what periodicities in the time series should be filtered

out prior to making a covariance estimate.

Our simulations showed that 10-day-integrated covariance estimates could broadly be considered as reliable for our 55 MHz multistatic radar configuration; shorter integration times may of course be possible for lower frequency radars with higher meteor detection rates. However, we did note that the uncertainty appears to asymptote towards a minimum value after about 10 days of integration; this value is clearly governed by the wave field characteristics. We also suggest that the accuracy and

precision of the covariance estimates may be able to be improved slightly by using a more rigorous radial velocity outlier rejection scheme than applied here.

*Code and data availability.* The simulation code developed in this study is available on request from Andrew J. Spargo, as is the data from the BP and Mylor meteor radars.





## Appendix A: Extraction of tidal features through the use of a wavelet transform

The time series reconstructed from the wavelet transform can be expressed as (Torrence and Compo (1998), eqn. (11)):

$$x_n = \frac{\delta j \, \delta t^{1/2}}{C_\delta \, \psi_0(0)} \sum_{j=0}^{J} \frac{\Re(W_n(s_j))}{s_j^{1/2}}, \tag{A1}$$

where $\delta j$ describes the wavelet scale separation, $\delta t$ represents the time separation between adjacent points, $J$ is the number of wavelet scales, $C_\delta$ is a reconstruction factor (0.776 for the Morlet wavelet), $\psi_0(0)$ is an energy scaling factor ($\pi^{-1/4}$ for the Morlet wavelet), $s_j$ are the wavelet scales, and $W_n(s_j)$ contains the complex wavelet transform coefficients at scale $s_j$. In reconstructing the hourly-averaged wind time series (regardless of the time series length), we have taken $\delta j = 0.02$, in contrast to Torrence and Compo (1998) (Sect. 2f), who choose $\delta j = 0.125$ in their example with the Morlet wavelet; we have done this to reduce the spacing between adjacent wavelet scales and hence improve the accuracy of the reconstruction. Also in contrast to Torrence and Compo (1998) (Sect. 2g), we have not applied any zero padding in the application of the wavelet transform. This was done given our finding that the magnitude of artefacts at the ends of the wind time series appeared to be larger with zero padding applied.

## Appendix B: SABER-derived density climatology creation

To create a climatology of the diurnal variability in density from SABER instrument data that was representative of conditions around Adelaide during the autumnal equinox, we acquired densities from individual limb scans with tangent point latitudes spanning 28°S-42°S, longitudes 108°E-168°E, days 01-March to 31-May inclusive, and years 2008-2018 inclusive. Measurements falling into given time-of-day (hourly) and height (0.5 km) bins were averaged.

A spatial sampling region and measurement time-of-year span of this size was necessary to fill all time-of-day bins with measurements. An average over 11 years of data was performed to reduce the level of aliasing arising from GW-induced perturbations occurring in individual scans.

The climatology produced using this method had features that were qualitatively consistent with the same time averaging on NRLMSISE-00 model output from Adelaide's location. However, we did note that given density surfaces from SABER were, on average, 2 km lower than NRLMSISE-00's predictions between about 80 and 95 km. Nevertheless, the use of the SABER-derived density climatology in the production of Fig. 13 yielded almost identical flow accelerations as the use of said NRLMSISE-00 output.

*Author contributions.* Andrew Spargo carried out the model development and data analysis, and wrote the paper. Iain Reid and Andrew MacKinnon are supervisors of Andrew Spargo's postgraduate candidature.



*Competing interests.* The meteor radars used in this study were designed and manufactured by ATRAD Pty. Ltd., and Iain Reid is the executive director of this group of companies.

*Acknowledgements.* Andrew Spargo would like to thank Jorge Chau, Chris Adami, Bob Vincent, David Holdsworth, Joel Younger, Richard Mayo, Andrew Heitmann, Yi Wen, Tom Chambers and Baden Gilbert for useful discussions regarding this work.

5     Andrew Spargo is supported by an Australian Government Research Training Program Scholarship.

The BP ST/Meteor radar is supported by ATRAD Pty. Ltd. and the University of Adelaide. The Mylor receiving site and equipment is supported solely by ATRAD Pty. Ltd.



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
