# Peer review of "Multistatic meteor radar observations of gravity wave-tidal interaction over Southern Australia"

_Atmospheric Measurement Techniques, 2019_

## Referee Comment (RC1) · Chris Meek (Referee) · 5 May 2019

[12pt]article

**Review of Multistatic meteor radar observations of gravity wave-tidal interaction over South Australia.**

**by Andrew J. Spargo et al.**

**( Atmos. Meas. Tech. Discuss. AMT-2019-138)**

[Figure]

This clearly written, very detailed paper, reports on a statistical simulation of bistatic meteor radar data including assumed realistic noise, biases, measurement errors, based on actual meteor distributions, and added pertubations from expected gravity wave parameters - the object being to estimate typical biases and uncertainties of e.g., $< u'w' >$, under various filtering and refection criteria.

A quite interesting observation is that the addition of a remote receiving site, which seemingly increases the diversity of meteor echoes does not contribute except by numbers: an expensive way to discover that, but valuable nonetheless.

Sporadic meteor distributions measured by radar are not uniform, e.g. in azimuth - they can vary hour to hour and day to day depending on the orientation of the source regions. This model is based on actual distributions. How much error in GW forcing could be due to non-uniform distributions, which are inherent in the use of meteor radar.

Pg 10 line 5: are the decay times for the remote site expected to be the same as for the main site?

Pg 11 line 21. Was the tidal phase adjusted for each meteor's position or is that correction judged to be overkill?

Pg 12 Line 5: I have brought this up before and been shot down - so I will try again. It appears that a basic assumption of the method is that the atmospheric motion at the meteor is perpendicular to the trail; that is, that the echoing region has a vertical

mitigate this concern to some extent.

Pg 12 line 10 - it's not clear what is meant by a (square?) radial/AOA pair - does that refer to a single meteor ?

Pg 12 Line 17 is there an extra "i" in this equation?

Figure 5,6 red(-dish) lines (and yellow) are almost invisible (despite the caption, I don't see any thick lines). Figure 7 is good (probably because all lines are thick - when some should be thin. Solid and dashed thick might look better.

Fig 13 The zero U and peak forcing 'line' are very close over the plotted heights. It appears that U leads the forcing. Curious.

Minor:

Pg. 6 line 20 "Spatially distributed..."?

—///—

---

## Referee Comment (RC2) · Chris Meek (Referee) · 9 May 2019

**Review of Multistatic meteor radar observations of gravity wave-tidal interaction over South Australia.**
**by Andrew J. Spargo et al.**
**( Atmos. Meas. Tech. Discuss. AMT-2019-138)**

This clearly written, very detailed paper, reports on a statistical simulation of bistatic meteor radar data including assumed realistic noise, biases, measurement errors, based on actual meteor distributions, and added pertubations from expected gravity wave parameters - the object being to estimate typical biases and uncertainties of e.g., $< u'w' >$, under various filtering and refection criteria.

A quite interesting observation is that the addition of a remote receiving site, which seemingly increases the diversity of meteor echoes does not contribute except by numbers: an expensive way to discover that, but valuable nonetheless.

Sporadic meteor distributions measured by radar are not uniform, e.g. in azimuth - they can vary hour to hour and day to day depending on the orientation of the source regions. This model is based on actual distributions. How much error in GW forcing could be due to non-uniform distributions, which are inherent in the use of meteor radar.

Pg 10 line 5: are the decay times for the remote site expected to be the same as for the main site?

Pg 11 line 21. Was the tidal phase adjusted for each meteor's position or is that correction judged to be overkill?

Pg 12 Line 5: I have brought this up before and been shot down - so I will try again. It appears that a basic assumption of the method is that the atmospheric motion at the meteor is perpendicular to the trail; that is, that the echoing region has a vertical velocity component. This might be true if there is a "hot spot" (point scatter) in the trail , but for a straight line reflector the reflection point would be expected to slide along the trail if necessary to maintain perpendicularity. There would be a very small change in zenith angle, but no vertical velocity is needed.

If there were a large numbers, uniform azimuthal/time meteor distribution at the height of interest, the sliding effect would be expected to have minimal influence. Otherwise significant covariances could be created from horizontal variations alone (no vertical motion).

Another question in the same vein: in the monostatic case, if there were only zonal wind perturbations, then because of the radial measurement, there would also appear to be meridional

perturbations. That is, zonal and meridional perturbations "bleed" into each other. Does this affect your results? It seems that bistatic operation would mitigate this concern to some extent.

Pg 12 line 10 - it's not clear what is meant by a (square?) radial/AOA pair - does that refer to a single meteor ?

Pg 12 Line 17 is there an extra "i" in this equation?

Figure 5,6 red(-dish) lines (and yellow) are almost invisible (despite the caption, I don't see any thick lines). Figure 7 is good (probably because all lines are thick - when some should be thin. Solid and dashed thick might look better.

Fig 13 The zero U and peak forcing 'line' are very close over the plotted heights. It appears that U leads the forcing. Curious.

Minor:
Pg. 6 line 20 "Spatially distributed..."?
$$—///—$$

---

## Referee Comment (RC3) · Anonymous Referee #2 · 6 Jun 2019

The paper represents an interesting study of momentum fluxes with a bistatic meteor radar. The spacing is quite small by normal standards (only 55 km) but the results are of interest. An increase in count rates is seen. However, a few items are not adequately explained or discussed, such as the importance of exclusion of overhead echoes (which determine primarily vertical velocities), and the proper calculation of Bragg vectors. Some references are in error, or have been ascribed undue originality.

Please also note the supplement to this comment:
https://www.atmos-meas-tech-discuss.net/amt-2019-138/amt-2019-138-RC3-supplement.pdf

[Figure]

**Supplement:**

Referee's report on the paper

Multistatic meteor radar observations of gravity wave-tidal interaction over Southern Australia

by Spargo, Reid and  MacKinnon

The paper looks at the accuracies of momentum flux determination using bistatic meteor radars, and some consequences. It is well written grammatically, and has been adequately proof-read. Indeed it was a pleasure for once to read a paper which did not require that I spend large amounts of extra time correcting grammar, so congratulations to the authors for that.

Scientifically, the paper is of interest but contains some significant points of potential confusion or even error. I shall address these in more detail later, but briefly these are:

1.  As shown in "Spatial distribution of errors associated with multistatic meteor radar", Earth, Planets and Space, 70:93, doi.org/10.1186/s40623-018-0860-2, 2018, some data need to be excluded from calculations, notably those close to the midpoint between the radar transmitter and receivers. In Fig 1., lower right, (bistatic case) these correspond to the region to the south-east of the transmitter, . While modest in number in this region, the radial velocities measured here relate mostly to the vertical velocities, and are best excluded for calculations of winds and momentum flux.  Failing to exclude these points can adversely affect the errors. For a monostatic system, these are all data close to overhead of the transmitter-receiver system (the blue-coloured cluster around the transmitter in the lower left figure). Typically data within 12-15 degrees of vertical are removed for the monostatic case (e.g. see Radio Sci., 32, 833-865, 1997).  A similar exclusion process (or at least a weighting which reduces the weight of these meteors) should be applied in this paper, but no mention of it is made.  While the numbers of meteors in these regions are low (in the blue colours), their effect can be disproportionate, so some discussion about how these cases are treated is warranted.

2. In the same spirit, Fig. 3 is a bad choice of figure, since it seems to refer to a horizontal meteor trail - which is the one case that should always be avoided, as it produces measurements of the vertical wind only, and consequently division by zero when trying to deduce "horizontal winds". I suggest using the  figure, but tilt it at some arbitrary (non-zero!) angle from the horizontal.

3.  The paper does not discuss how the Bragg angles are found. The paper by Stober et al. (2018) suggests they always point exactly to the centre-point of the transmitter-receiver line, but this is in error, and only approximately true (as some simple geometric calculations using high-school-level trigonometry will show). I believe the authors of the Stober et al. paper sent a correction to the journal in this regard. Incorrect calculation of the Bragg angle will bias the results.

4. The appendices (software) of Stober et al. contain couple of  typos - the authors should confirm that they found these.

5. The authors use a spherical-Earth calculation for the bistatic case, as they should. But it is unclear as to whether they did the same in the monostatic case. In some radars, a flat-Earth approximation is used for the monostatic case - the error in height is typically 0.5 km or so at

zenith angles of 45 degrees. But it is unclear here whether the flat-Earth approximation was used in the monostatic case, or whether a full spherical Earth approximation was used in both cases. If a flat-earth approximation is used in one case, and a spherical-Earth system used in another, it could lead to biases in comparison. This may also impact the upper-right graph in Fig. 1 (please add labels a,b,c,d - its getting annoying referring to "upper left", "lower right" etc.)

The above 5 points could bias the calculation of momentum fluxes. Perhaps the authors have considered these points, and simply not mentioned them, or perhaps they neglected these matters - either way, these matters must be considered and discussed in the paper.

Another point of note, though maybe less important, is the mean Bragg wavelength. The authors note that the bistatic procedure gives rise to a diversity of Bragg scales, but it must be remembered bistatic studies also alter the mean Bragg values. The effective Fourier scale involved in the reflection process is of course $D = \lambda/[2 \cos (\beta/2)]$ (using the notation of Fig 3). If $\beta/2$ is say $30^o$, $D = 0.58 \lambda$, rather than $0.5 \lambda$ as for the monostatic case. So it is as if the radar had a frequency of 47 MHz rather than 55 MHz. So it might be of interest to note that the mean effective frequency has been lowered - and meteor detections tend to be more common at frequencies closer to 30 MHz. The improvement in detection rated may be more significant in the case of a 55 MHz radar because 55 MHz is not normally considered optimum for meteor work, so a shift to 47 MHz might be more dramatic (relatively). It would be of interest for the authors to look not just at diversity effects but also to consider the mean effective wavelength due to the bistatic arrangement. My choice of $\beta$ of 60 degrees is arbitrary - the authors should be able to find a more realistic mean using averaging over real values of $\beta$.

Now I will progress to other matters, and work through page-by-page.

Page 6, line 10 - the term "said distributions" seems clumsy - the word "said" is used in this way several times - it does not add anything to the text, and I would recommend removing it.

Page 8, line 4 - the authors talk about "well-known linear GW polarization relations". There is no *single set* of "linear relations" -this is just one of many, and it is the simplest. It all depends on whether you include Coriolis terms, Brunt-Vaisala terms, scale-height terms etc -these cases *still deal with linear waves*, but the detailed formulas change. See Walterscheid et al., JAS, "Stokes diffusion by atmospheric internal gravity waves", *J. Atmos. Sci.*, **48**, 2213-2230, 1991, equation (45) and set $\alpha$ to 0 for a more general linearized case.

So this misleading statement needs to be clarified, perhaps by saying "we use the simplest polarization relations". Of course it needs to be recognized that the *authors own work will be in error at periods close to the tidal and inertial frequencies if they use this simple polarization relation* - Coriolis terms should be included for a proper treatment, for example. This is pertinent to discussions later in the text, where the authors consider the optimum bandwidth for studies and look at GW periods close to the tides.

Page 9, line 9 --- I would make the "dot product" between $\mathbf{v}_{ecef}$ and $\mathbf{e}$ clearer - maybe use a bold dot - it is important that readers do not miss this, and plays into the discussions introduced earlier regarding overhead meteors. The equation emphasizes that when the meteors are close to overhead, the radial velocity is dominated by the vertical wind component (also Hocking, Earth,

Planets and Space, 70:93, doi.org/10.1186/s40623-018-0860-2, 2018 ( + see item 1 on page 1 of this review). As noted, it would be a good idea to change Fig. 3 to have a generalized tilt for the meteor.

Incidentally, the authors used bold font to represent both vectors and matrices. While not wrong, since a vector is indeed a type of matrix, they may want to consider distinguishing these cases e.g. by using an arrow over the vectors and keeping matrices bold.

Page 10, line 18 - how "rare" is "rare"?

Page 10, lines 19 to 21, and page 11, lines 1 - 14.
    These lines are very disturbing. First, the authors seem to have the wrong Holdsworth et al. (2004) paper in the References.  The authors give Holdsworth, D. A., Tsutsumi, M., Reid, I. M., Nakamura, T., and Tsuda, T.: Interferometric meteor radar phase calibration using meteor echoes, Radio Sci., 39, 2004. Yet line 19 on page 10 discusses an "error code 3".  I cannot see an "error code 3" in this paper. I CAN find one in the paper "Holdsworth, D. A., I. M. Reid, and M. A. Cervera (2004), Buckland Park all-sky interferometric meteor radar, Radio Sci., 39, RS5009, doi:10.1029/2003RS003014". It appears that the authors have referenced the wrong paper, and I believe this is true for (almost?) all references to Holdsworth et al., (2004). Please check. I should not need to have to double-check your own references!
But it gets worse. On page 11, line 3, the authors give Holdsworth (2004) as a representative example of "standard analysis", and then on line 11, make the statement "...  we follow the iterative scheme PROPOSED by Holdsworth et al., (2004).".  Sorry ,but this is very misleading. This iterative scheme was proposed fully *7 years earlier* (Hocking and Thayaparan, Radio Sci., 32, 833-865, 1997, section 4.1) and again discussed in Hocking et al., *J. Atmos. Solar-Terr. Physics*, **63**, 155-169, 2001.). Holdsworth may have made minor adjustments, but he did not "propose" the idea - it was already proposed 7 years earlier. Indeed much of Table 2 in Holdsworth et al (2004) was also proposed 3 years before Holdsworth (2004) - in Hocking et al., (2001). Repeated reference to Holdsworth throughout the text in regard to ideas which were proposed many years earlier by others is unprofessional and misleading. Perhaps the first author (who, I believe seems to have written much of the paper - and written it quite well, I will add) was simply unaware of these much earlier works, but a proper historical perspective must be presented by rightfully referencing these earlier works in a suitable manner.

In regard to sections 3.7 and on - I again ask - were cases close to the zenith, where the radial velocity is dominated by vertical winds, removed, or given lower weighting? Or how were such cases dealt with?  Nasty near-infinity terms can occur when such terms are include, which could bias the resultant mean winds and momentum fluxes.

Page 12, lines 1 to 9. This is another example of poor citations, perhaps due to the first author's inexperience. Line 1 says "The approach we apply is identical to that presented by Thorsen et al., (1997) and subsequently Hocking (2005)."  It gives the impression that the ideas were developed by Thorsen and simply re-applied by Hocking. This is patently untrue. The current authors use the matrix **A'** as discussed in lines 7 to 8.  Thorsen et al. NEVER used this matrix. The matrix Thorsen et al. used is given in their equation (18), section 3.3, page 714. It is a MUCH simplified version of the true **A'**. The matrix that the current authors use comes straight out of Hocking

(2004) and was never used by Thorsen et al. It makes a major improvement to the analysis. So please replace "subsequently" in line 1 of page 1 with "subsequently improved", and give a specific reference to Hocking (2004) in regard to lines 7 and 8.

Page 12, line 26 - "we have opted to compute the covariances at the origin" - do the authors mean "at the origin at the ground" or "at a point in the atmosphere at meteor heights immediately above the ground". I can see that either might work, but it needs to be clarified.

Page 13, line 1. The statement that "This estimate represents what one would measure with an "anemometer" at some fixed location in the vicinity of the radar", is a little misleading. I understand what the authors are trying to say, but it isn't quite right, and needs some caveats added. First, "in the vicinity of the radar" is misleading. Do the authors mean "in the vicinity of the meteor region immediately above the radar"? As it reads, it could be at ground-level. In addition, in the bistatic case, which radar? the transmitter? or the receivers? True, in this case they are moderately close, but since the report is supposed to somehow represent general multistatic radars, where the transmitter and receiver can be hundreds of km apart, the question is no longer pedantic. Furthermore, a suitable anenometer might truly measure wind components, where with the radar we have this problem that we deal with radial components. So while I get what the author is trying to say, I suggest it be mentioned with some more caution than has been applied..

Page 14, lines 2-3 - good point.

Page 14, lines 9-10 - OK, but as discussed earlier, could some of the increase in count rates be due to a new effective mean frequency? And I also ask again - did the monostatic radar use the same "spherical Earth" geometry as the bistatic case?

Pages 15-21 seem quite good. However, the term "gap" is used in Figs. 10, 11, in line 10, page 21. "Gap" gives the impression of a "hole" - I would say the term "temporal shift" might be more appropriate.

Page 21, line 1 - the authors say ".. aside from considering that the GWs may have propagated from a region with weak eastward mesospheric winds." Do they really mean this? Or do they mean "... GWs may have propagated through a region with weak eastward..."? It's the filtering that matters - so either the statement needs some expansion, or the authors really mean "through".

Page 23 line 2 - give a reference to the forcing formula.

Page 23, line 28 - suggest changing "Also like.." to "As for..."

Page 24, lines 11-12  - seems a bit unclear - please try to reword what you mean.

Pages 26 to 28 are a nice summary of the literature, and really emphasize current uncertainties and confusion in this regard.

Page 28, conclusions. Some of these statements may need revisiting following other comments discussed here-in.

============= End ==================

---

## Author Comment (AC2) · 18 Jun 2019

**Author's response to Anonymous Referee #2's comments (amt-2019-138)**

Last updated: Tuesday  $18^{\rm th}$  June, 2019 at 07:34 UT

The authors kindly thank the anonymous reviewer for their comprehensive review of the paper. We believe that the suggestions made that we have been able to address have significantly improved the paper. Below are our responses to the points raised, indicating the changes made to the manuscript where appropriate.

| No. | Referee's comment                                  | Author's response and changes                       |
|-----|----------------------------------------------------|-----------------------------------------------------|
|     | 1. As shown in "Spatial distribution of errors     |                                                     |
|     | associated with multistatic meteor radar", Earth,  |                                                     |
|     | Planets and Space, 70:93,                          |                                                     |
|     | doi.org/10.1186/s40623-018-0860-2, 2018, some      |                                                     |
|     | data need to be excluded from calculations,        | We acknowledge that horizontal wind/momentum        |
|     | notably those close to the midpoint between the    | flux estimation errors arising from radial velocity |
|     | radar transmitter and receivers. In Fig 1., lower  | measurement errors are amplified for meteors        |
|     | right, (bistatic case) these correspond to the     | close to the transmitter-receiver midpoint.         |
|     | region to the south-east of the transmitter, .     | However, as the reviewer alludes to, there are so   |
|     | While modest in number in this region, the radial  | few meteors detected here (less than around $10\%$  |
|     | velocities measured here relate mostly to the      | per unit area than at the peaks of the distribution |
|     | vertical velocities, and are best excluded for     | in our case), that we did not consider filtering    |
|     | calculations of winds and momentum flux. Failing   | them from the analysis or weighting them in any     |
|     | to exclude these points can adversely affect the   | way. We believe that weighting the meteor           |
| 1.  | errors. For a monostatic system, these are all     | distribution to minimize the errors in the fitted   |
|     | data close to overhead of the transmitter-receiver | wind/momentum fluxes is a separate topic            |
|     | system (the blue-coloured cluster around the       | warranting its own investigation, and so we have    |
|     | transmitter in the lower left figure). Typically   | added a comment to this effect to the discussion    |
|     | data within 12-15 degrees of vertical are removed  | section of the manuscript. Nevertheless, the        |
|     | for the monostatic case (e.g. see Radio Sci., 32,  | simulations performed in this paper also do not     |
|     | 833-865, 1997). A similar exclusion process (or at | incorporate any weighting of the meteors in the     |
|     | least a weighting which reduces the weight of      | fits, so the momentum flux errors we have quoted    |
|     | these meteors) should be applied in this paper,    | are still relevant for the analysis we performed on |
|     | but no mention of it is made. While the numbers    | the real data.                                      |
|     | of meteors in these regions are low (in the blue   |                                                     |
|     | colours), their effect can be disproportionate, so |                                                     |
|     | some discussion about how these cases are treated  |                                                     |
|     | is warranted.                                      |                                                     |
|     | 2. In the same spirit, Fig. 3 is a bad choice of   |                                                     |
|     | figure, since it seems to refer to a horizontal    |                                                     |
|     | meteor trail - which is the one case that should   |                                                     |
|     | always be avoided, as it produces measurements     |                                                     |
| 2.  | of the vertical wind only, and consequently        | Agreed, modified as suggested.                      |
|     | division by zero when trying to deduce             |                                                     |
|     | "horizontal winds". I suggest using the figure,    |                                                     |
|     | but tilt it at some arbitrary (non-zero!) angle    |                                                     |
|     | from the horizontal.                               |                                                     |

| 3. | 3. The paper does not discuss how the Bragg
angles are found. The paper by Stober et al.
(2018) suggests they always point exactly to the
centre-point of the transmitter-receiver line, but
this is in error, and only approximately true (as
some simple geometric calculations using
high-school-level trigonometry will show). I
believe the authors of the Stober et al. paper sent
a correction to the journal in this regard.
Incorrect calculation of the Bragg angle will bias
the results.                                                                                                                                                                                                    | By "Bragg angle", we assume the reviewer is
referring to the direction of the vector that is
perpendicular to the meteor trail. In this paper,
we have assumed this is in the direction of the
bisector of the vectors from the meteor to the
transmitter and the meteor to the receiver (which
have been referred to as $t$ and $b$ , respectively),
and the way this has been estimated has already
been discussed in Sect. 3.6. We agree with the
reviewer's point that this vector does not
necessarily point towards the midpoint between
the transmitter and receiver, as suggested in
Stober et al. (2018), Fig. 9.
As part of this correction, we have made some
modifications to how we discuss the use of the
Bragg angle/vector in the section on wind (3.7)
and covariance (3.9) estimation. |
|----|---------------------------------------------------------------------------------------------------------------------------------------------------------------------------------------------------------------------------------------------------------------------------------------------------------------------------------------------------------------------------------------------------------------------------------------------------------------------------------------------------------------------------------------------------------------------------------------------------------------------------------------------------------------------------------------------------------------------------------------|-----------------------------------------------------------------------------------------------------------------------------------------------------------------------------------------------------------------------------------------------------------------------------------------------------------------------------------------------------------------------------------------------------------------------------------------------------------------------------------------------------------------------------------------------------------------------------------------------------------------------------------------------------------------------------------------------------------------------------------------------------------------------------------------------------------------------------------------------------------------|
| 4. | 4. The appendices (software) of Stober et al.
contain couple of typos - the authors should
confirm that they found these.                                                                                                                                                                                                                                                                                                                                                                                                                                                                                                                                                                                                       | We found one typo in the appendices of this
paper, and have now made note of this in the
manuscript.                                                                                                                                                                                                                                                                                                                                                                                                                                                                                                                                                                                                                                                                                                                                                      |
| 5. | 5. The authors use a spherical-Earth calculation
for the bistatic case, as they should. But it is
unclear as to whether they did the same in the
monostatic case. In some radars, a flat-Earth
approximation is used for the monostatic case -
the error in height is typically 0.5 km or so at
zenith angles of 45 degrees. But it is unclear here
whether the flat-Earth approximation was used in
the monostatic case, or whether a full spherical
Earth approximation was used in both cases. If a
flat-earth approximation is used in one case, and
a spherical-Earth system used in another, it could
lead to biases in comparison. This may also
impact the upper-right graph in Fig. 1 | We have added comments in several places to
clarify that we assumed an ellipsoidal Earth for
both bistatic and monostatic cases.                                                                                                                                                                                                                                                                                                                                                                                                                                                                                                                                                                                                                                                                                                                          |
| 6. | please add labels a,b,c,d - its getting annoying
referring to "upper left", "lower right" etc.                                                                                                                                                                                                                                                                                                                                                                                                                                                                                                                                                                                                                                     | Added labels to all figures where appropriate.                                                                                                                                                                                                                                                                                                                                                                                                                                                                                                                                                                                                                                                                                                                                                                                                                  |
| 7. | Another point of note, though maybe less
important, is the mean Bragg wavelength. The
authors note that the bistatic procedure gives rise
to a diversity of Bragg scales, but it must be
work, so a shift to 47 MHz might be more
dramatic (relatively) It would be of interest
for the authors to look not just at diversity effects
but also to consider the mean effective wavelength
due to the bistatic arrangement.                                                                                                                                                                                                                                                                                     | This is a good point—we have added a histogram
of the effective radar frequency to the paper.                                                                                                                                                                                                                                                                                                                                                                                                                                                                                                                                                                                                                                                                                                                                                                |

|     | Page 6, line 10 - the term "said distributions"       |                                                      |  |
|-----|-------------------------------------------------------|------------------------------------------------------|--|
| 8.  | seems clumsy - the word "said" is used in this        |                                                      |  |
|     | way several times - it does not add anything to       | Removed.                                             |  |
|     | the text, and I would recommend removing it.          |                                                      |  |
|     | Page 8, line 4 - the authors talk about               |                                                      |  |
|     | "well-known linear GW polarization relations".        |                                                      |  |
|     | There is no single set of "linear relations" -this is |                                                      |  |
|     | just one of many, and it is the simplest. It all      |                                                      |  |
|     | depends on whether you include Coriolis terms,        |                                                      |  |
|     | Brunt-Vaisala terms, scale-height terms etc -these    |                                                      |  |
|     | cases still deal with linear waves, but the detailed  | Thank you for raising this important point. We       |  |
|     | formulas change So this misleading statement          | have removed the "well-known" appellation. In        |  |
|     | needs to be clarified, perhaps by saying "we use      | addition, we only use wave periods up to 4 hours     |  |
| 9.  | the simplest polarization relations". Of course it    | in the simulations, which is well below the inertial |  |
|     | needs to be recognized that the authors own work      | and predominant tidal periods. Therefore, we         |  |
|     | will be in error at periods close to the tidal and    | believe it is unnecessary to qualify the potential   |  |
|     | inertial frequencies if they use this simple          | lack of realism associated with using the simpler    |  |
|     | polarization relation - Coriolis terms should be      | polarization relations.                              |  |
|     | included for a proper treatment, for example.         |                                                      |  |
|     | This is pertinent to discussions later in the text,   |                                                      |  |
|     | where the authors consider the optimum                |                                                      |  |
|     | bandwidth for studies and look at GW periods          |                                                      |  |
|     | close to the tides.                                   |                                                      |  |
|     | Page 9, line 9 — I would make the "dot product"       |                                                      |  |
|     | between vecef and e clearer - maybe use a bold        |                                                      |  |
|     | dot - it is important that readers do not miss this,  |                                                      |  |
| 10  | and plays into the discussions introduced earlier     | A much dama                                          |  |
| 10. | regarding overhead meteors. The equation              | Agreed, done.                                        |  |
|     | emphasizes that when the meteors are close to         |                                                      |  |
|     | overhead, the radial velocity is dominated by the     |                                                      |  |
|     | vertical wind component                               |                                                      |  |
|     | Incidentally, the authors used bold font to           |                                                      |  |
| 11. | represent both vectors and matrices. While not        | We have tried doing this, but the Copernicus         |  |
|     | wrong, since a vector is indeed a type of matrix,     | LaTeX class does not permit arrows over vectors.     |  |
|     | they may want to consider distinguishing these        | All vectors have been changed to bold/italicized     |  |
|     | cases e.g. by using an arrow over the vectors and     | to distinguish them from matrices (bold).            |  |
|     | keeping matrices bold.                                |                                                      |  |
| 19  | Page 10, line 18 - how "rare" is "rare"?              | Thank you for picking up on this—we have             |  |
| 12. |                                                       | re-examined and clarified accordingly.               |  |

|     | Page 10, lines 19 to 21, and page 11, lines 1 - 14.                    |                                                   |  |
|-----|------------------------------------------------------------------------|---------------------------------------------------|--|
|     | These lines are very disturbing. First, the authors                    |                                                   |  |
|     | seem to have the wrong Holdsworth et al. $(2004)$                      |                                                   |  |
|     | paper in the References. The authors give                              |                                                   |  |
|     | Holdsworth, D. A., Tsutsumi, M., Reid, I. M.,                          |                                                   |  |
|     | Nakamura, T., and Tsuda, T.: Interferometric                           |                                                   |  |
|     | meteor radar phase calibration using meteor                            |                                                   |  |
|     | echoes, Radio Sci., 39, 2004. Yet line 19 on page                      | Thank you for picking up on the incorrect         |  |
| 13. | 10 discusses an "error code 3". I cannot see an                        | reference—this was a mistake on our part. This    |  |
|     | "error code 3" in this paper. I CAN find one in                        | has been corrected.                               |  |
|     | the paper "Holdsworth, D. A., I. M. Reid, and M.                       |                                                   |  |
|     | A. Cervera (2004), Buckland Park all-sky                               |                                                   |  |
|     | interferometric meteor radar, Radio Sci., 39,                          |                                                   |  |
|     | RS5009, doi:10.1029/2003RS003014". It appears                          |                                                   |  |
|     | that the authors have referenced the wrong                             |                                                   |  |
|     | paper, and I believe this is true for (almost?) all                    |                                                   |  |
|     | references to Holdsworth et al., (2004).                               |                                                   |  |
|     | But it gets worse. On page 11, line 3, the authors                     |                                                   |  |
|     | give Holdsworth $(2004)$ as a representative                           |                                                   |  |
|     | example of "standard analysis", and then on line                       |                                                   |  |
|     | 11, make the statement " we follow the                                 |                                                   |  |
|     | iterative scheme PROPOSED by Holdsworth et                             |                                                   |  |
|     | al., (2004).". Sorry ,
but this is very misleading.                 |                                                   |  |
|     | This iterative scheme was proposed fully 7 years                       |                                                   |  |
|     | earlier (Hocking and Thayaparan, Radio Sci., 32,                       |                                                   |  |
|     | $833\text{-}865,1997,\mathrm{section}$ 4.1) and again discussed in     | We enclosing for missing this reference, this has |  |
| 14. | Hocking et al., J. Atmos. Solar-Terr. Physics, 63,                     | we apologize for missing this reference—this has  |  |
|     | 155-169, 2001.). Holds
worth may have made                          | now been added.                                   |  |
|     | minor adjustments, but he did not "propose" the                        |                                                   |  |
|     | idea - it was already proposed 7 years earlier.                        |                                                   |  |
|     | Indeed much of Table 2 in Holdsworth et al $(2004)$                    |                                                   |  |
|     | was also proposed 3 years before Holdsworth                            |                                                   |  |
|     | $\left(2004\right)$ - in Hocking et al., $\left(2001\right).$ Repeated |                                                   |  |
|     | reference to Holdsworth throughout the text in                         |                                                   |  |
|     | regard to ideas which were proposed many years                         |                                                   |  |
|     | earlier by others is unprofessional and misleading.                    |                                                   |  |
|     |                                                                        | As discussed earlier in this reply, we did not    |  |
|     | In regard to sections 3.7 and on - I again as
k -                   | consider removing/weighting the meteors in this   |  |
|     | were cases close to the zenith, where the radial                       | region given the small number of detections here. |  |
|     | velocity is dominated by vertical winds, removed,                      | Again, we believe that weighting the meteor       |  |
| 15. | or given lower weighting? Or how were such cases                       | distribution to minimize the wind/momentum        |  |
|     | dealt with? Nasty near-infinity terms can occur                        | flux estimation errors is a separate topic        |  |
|     | when such terms are include, which could bias the                      | warranting its own investigation, and have        |  |
|     | resultant mean winds and momentum fluxes.                              | mentioned this in the discussion section of the   |  |
|     |                                                                        | manuscript.                                       |  |

|     | Page 12, lines 1 to 9. This is another example of   |                                                     |
|-----|-----------------------------------------------------|-----------------------------------------------------|
|     | poor citations, perhaps due to the first author's   |                                                     |
|     | inexperience. Line 1 says "The approach we apply    |                                                     |
|     | is identical to that presented by Thorsen et al.,   | TT 1 1 . 1.1                                        |
|     | (1997) and subsequently Hocking (2005)." It gives   | We have adjusted this section to avoid implying     |
|     | the impression that the ideas were developed by     | that Hocking (2005) simply re-applied the           |
|     | Thorsen and simply re-applied by Hocking. This      | estimator shown in Thorsen et al. (1997).           |
|     | is patently untrue. The current authors use the     | However, it should be noted that the two            |
|     | matrix A' as discussed in lines 7 to 8. Thorsen et  | approaches are fundamentally the same; the          |
|     | al. NEVER used this matrix. The matrix              | Hocking $(2005)$ paper is just more descriptive in  |
| 16. | Thorsen et al. used is given in their equation      | that it outlines a way of inverting the A' matrix   |
|     | (18), section 3.3, page 714. It is a MUCH           | (by directly finding the least squares solution to  |
|     | simplified version of the true A' The matrix that   | the problem). We also note in this paper that we    |
|     | the current authors use comes straight out of       | do not use the inversion method described by        |
|     | Hocking $(2004)$ and was never used by Thorsen et   | Hocking (2005), but rather SVD, due to the fact     |
|     | a) It makes a major improvement to the analysis     | that SVD is known to provide more reliable          |
|     | So please replace "subsequently" in line 1 of page  | solutions to poorly-conditioned problems.           |
|     | 1 with "subsequently improved" and give a           |                                                     |
|     | specific reference to Hocking $(2004)$ in regard to |                                                     |
|     | lines 7 and 8                                       |                                                     |
|     | Page 12 line 26 "we have opted to compute the       |                                                     |
|     | age 12, line 20 - we have opted to compute the      |                                                     |
|     | "at the origin at the ground" or "at a point in the |                                                     |
| 17. | at the origin at the ground of at a point in the    | This has now been clarified.                        |
|     | the ground". Lean see that either might work        |                                                     |
|     | but it poods to be clarified                        |                                                     |
|     | Page 12 line 1. The statement that "This            |                                                     |
|     | rage 13, line 1. The statement that This            |                                                     |
|     | estimate represents what one would measure with     |                                                     |
|     | an anemometer at some fixed location in the         |                                                     |
|     | vicinity of the radar, is a little misleading. I    | We have removed the statements about the            |
|     | but it ion't quite right and peads some sevents     | anemometer and about the reference being "in        |
|     | added Einst "in the minimity of the reder" is       | the vicinity of the radar". We have already briefly |
|     | micloading. Do the authors mean "in the vicinity    | discussed the point about the correlation of the    |
| 10  | of the meteor perior immediately above the          | wave field over the entire observation volume in    |
| 10. | of the meteor region immediately above the          | Sect. 3.10; we found the covariances evaluated at   |
|     | radar ? As it reads, it could be at ground-level.   | the coordinate system origin vs. at the             |
|     | in addition, in the distance case, which radar? the | positions/times of the meteors to be in very close  |
|     | than and me denote by the bet in this case          | agreement. On the basis of this, we assumed use     |
|     | they are moderately close, but since the report is  | of the "origin" reference point to be valid.        |
|     | supposed to somenow represent general               |                                                     |
|     | multistatic radars, where the transmitter and       |                                                     |
|     | receiver can be hundreds of km apart, the           |                                                     |
|     | question is no longer pedantic.                     |                                                     |

|     |                                                     | I do not understand the specific problem the        |  |
|-----|-----------------------------------------------------|-----------------------------------------------------|--|
| 19. | Furthermore, a suitable anenometer might truly      | reviewer is discussing here. In any case, the whole |  |
|     | measure wind components, where with the radar       | premise of the simulation has been to estimate      |  |
|     | we have this problem that we deal with radial       | the errors associated with propagating realistic    |  |
|     | components. So while I get what the author is       | radial velocity errors to the covariance estimates; |  |
|     | trying to say, I suggest it be mentioned with some  | the "perfect anemometer" winds, assuming the        |  |
|     | more caution than has been applied.                 | radar is attempting to measure these, are used as   |  |
|     | T T                                                 | the reference for estimating the size of the error. |  |
|     |                                                     | Thank you for picking up on the lack of clarity     |  |
|     |                                                     | have we have reworded this sentence to make it      |  |
|     |                                                     | clean that we are talling chaut the increase in     |  |
|     |                                                     | clear that we are taiking about the increase in     |  |
|     |                                                     | counts associated with combining the monostatic     |  |
|     | Page 14, lines 9-10 - OK, but as discussed earlier, | and bistatic receiver data. There is indeed a       |  |
|     | could some of the increase in count rates be due    | slight increase in counts at the higher altitudes   |  |
| 20. | to a new effective mean frequency? And I also       | due to the effective lower operating frequency, but |  |
|     | ask again - did the monostatic radar use the same   | this is not related to the point being made, and    |  |
|     | "spherical Earth" geometry as the bistatic case?    | has already been mentioned in Sect 2.1. As          |  |
|     |                                                     | discussed earlier in this reply, we also used the   |  |
|     |                                                     | same ellipsoidal Earth model to compute             |  |
|     |                                                     | meteor-related parameters in the monostatic and     |  |
|     |                                                     | bistatic cases.                                     |  |
|     | Pages 15-21 seem quite good. However, the term      |                                                     |  |
|     | "gap" is used in Figs. 10, 11, in line 10, page 21. |                                                     |  |
| 21. | "Gap" gives the impression of a "hole" - I would    | Agreed, made suggested change.                      |  |
|     | say the term "temporal shift" might be more         |                                                     |  |
|     | appropriate.                                        |                                                     |  |
|     | Page 21, line 1 - the authors say " aside from      |                                                     |  |
|     | considering that the GWs may have propagated        |                                                     |  |
|     | from a region with weak eastward mesospheric        |                                                     |  |
|     | winds." Do they really mean this? Or do they        | Thank you for picking up on this—changed to         |  |
| 22. | mean " GWs may have propagated through a            | "through" as suggested.                             |  |
|     | region with weak eastward "? It's the filtering     |                                                     |  |
|     | that matters - so either the statement needs some   |                                                     |  |
|     | expansion or the authors really mean "through"      |                                                     |  |
|     | Page 23 line 2 give a reference to the forcing      |                                                     |  |
| 23. | formula                                             | Done.                                               |  |
|     |                                                     |                                                     |  |
| 24. | Page 23, line 28 - suggest changing "Also like"     | Done.                                               |  |
|     | to "As for"                                         |                                                     |  |
| 25. | Page 24, lines 11-12 - seems a bit unclear - please | Thank you for picking up on this lack of            |  |
|     | try to reword what you mean.                        | clarity—modified.                                   |  |
|     | Page 28, conclusions. Some of these statements      | None of the modifications affect the conclusion     |  |
| 26. | may need revisiting following other comments        | section, so we have left it unchanged.              |  |
|     | discussed here-in.                                  | seeden, so no have for it dichailged.               |  |

[revised manuscript text omitted]

- 10

5

---

## Author Response (AR1)

**Author's response to Chris Meek's comments (amt-2019-138)**

Last updated: Tuesday 11th June, 2019 at 07:53 UT

The authors kindly thank Chris Meek for his comments. Below are our responses to the comments, indicating the changes made to the manuscript where appropriate.

| No. | Referee's comment | Author's response and changes |
|---|---|---|
| 1. | Pg 10 line 5: are the decay times for the remote site expected to be the same as for the main site? | This is a good point; the decay time distribution for the bistatic receiver will have a slightly larger width than that of the monostatic receiver, given the distribution of Bragg wavelengths in the former case. However, we do not expect this to significantly change the parameters estimated from the synthesized time series, and so accordingly we have not made any changes to the manuscript. |
| 2. | Pg 11 line 21. Was the tidal phase adjusted for each meteor's position or is that correction judged to be overkill? | Yes, the horizontal wind time series was interpolated to the time of each meteor. Thank you for raising this point - we have included it in the manuscript. |
| 3. | Pg 12 Line 5: I have brought this up before and been shot down - so I will try again. It appears that a basic assumption of the method is that the atmospheric motion at the meteor is perpendicular to the trail; that is, that the echoing region has a vertical velocity component. This might be true if there is a hot spot (point scatter) in the trail , but for a straight line reflector the reflection point would be expected to slide along the trail if necessary to maintain perpendicularity. There would be a very small change in zenith angle, but no vertical velocity is needed. If there were a large numbers, uniform azimuthal/time meteor distribution at the height of interest, the sliding effect would be expected to have minimal influence. Otherwise significant covariances could be created from horizontal variations alone (no vertical motion). | As the reviewer indicates, this idea was raised in a manuscript that was rejected from publication on the basis of it not providing any observational evidence of scattering points moving along meteor trails. We consider it inappropriate to discuss a previously rejected idea in our manuscript, in the absence of any supporting evidence of it. Nevertheless, we acknowledge that the source of large biases in meteor radar-derived vertical wind estimates remains to be determined. |

| | | |
|---|---|---|
| 4. | Another question in the same vein: in the monostatic case, if there were only zonal wind perturbations, then because of the radial measurement, there would also appear to be meridional perturbations. That is, zonal and meridional perturbations "bleed" into each other. Does this affect your results? It seems that bistatic operation would mitigate this concern to some extent. | As above, we are not able to comment on this idea until observational evidence of it is provided. |
| 5. | Pg 12 line 10 - it's not clear what is meant by a (square?) radial/AOA pair - does that refer to a single meteor ? | Yes, this does refer to a single meteor. Thank you for spotting this lack of clarity — have modified manuscript accordingly. |
| 6. | Pg 12 Line 17 is there an extra i in this equation? | Yes — corrected. |
| 7. | Figure 5,6 red(-dish) lines (and yellow) are almost invisible (despite the caption, I dont see any thick lines). Figure 7 is good (probably because all lines are thick - when some should be thin. Solid and dashed thick might look better. | We appreciate the concern. We originally opted to use dashed lines in place of the thinner lines, but found these to be even more difficult to view. Therefore, we have left the figure as is. |
| 8. | Fig 13 The zero U and peak forcing line are very close over the plotted heights. It appears that U leads the forcing. Curious. | Indeed. These are both prospects for further investigation. Were we to mention the coincidence of the zero $u$ and peak $F_x$, we would also need to discuss how this finding fits in with previous similar investigations. Since it is beyond the scope of the present paper, which is to discuss the phase differences between the tidal winds and flow accelerations, we have decided to omit this point. The point concerning the phase relationship of $u$ and $F_x$ has also already been discussed in Sect 4.2. |
| 9. | Pg. 6 line 20 Spatially distributed...? | Good point, we have modified this accordingly. |

**Author's response to Anonymous Referee #2's comments (amt-2019-138)**

Last updated: Tuesday 18[th] June, 2019 at 07:34 UT

The authors kindly thank the anonymous reviewer for their comprehensive review of the paper. We believe that the suggestions made that we have been able to address have significantly improved the paper. Below are our responses to the points raised, indicating the changes made to the manuscript where appropriate.

| No. | Referee's comment | Author's response and changes |
|---|---|---|
| 1. | 1. As shown in "Spatial distribution of errors associated with multistatic meteor radar", Earth, Planets and Space, 70:93, doi.org/10.1186/s40623-018-0860-2, 2018, some data need to be excluded from calculations, notably those close to the midpoint between the radar transmitter and receivers. In Fig 1., lower right, (bistatic case) these correspond to the region to the south-east of the transmitter, . While modest in number in this region, the radial velocities measured here relate mostly to the vertical velocities, and are best excluded for calculations of winds and momentum flux. Failing to exclude these points can adversely affect the errors. For a monostatic system, these are all data close to overhead of the transmitter-receiver system (the blue-coloured cluster around the transmitter in the lower left figure). Typically data within 12-15 degrees of vertical are removed for the monostatic case (e.g. see Radio Sci., 32, 833-865, 1997). A similar exclusion process (or at least a weighting which reduces the weight of these meteors) should be applied in this paper, but no mention of it is made. While the numbers of meteors in these regions are low (in the blue colours), their effect can be disproportionate, so some discussion about how these cases are treated is warranted. | We acknowledge that horizontal wind/momentum flux estimation errors arising from radial velocity measurement errors are amplified for meteors close to the transmitter-receiver midpoint. However, as the reviewer alludes to, there are so few meteors detected here (less than around 10% per unit area than at the peaks of the distribution in our case), that we did not consider filtering them from the analysis or weighting them in any way. We believe that weighting the meteor distribution to minimize the errors in the fitted wind/momentum fluxes is a separate topic warranting its own investigation, and so we have added a comment to this effect to the discussion section of the manuscript. Nevertheless, the simulations performed in this paper also do not incorporate any weighting of the meteors in the fits, so the momentum flux errors we have quoted are still relevant for the analysis we performed on the real data. |
| 2. | 2. In the same spirit, Fig. 3 is a bad choice of figure, since it seems to refer to a horizontal meteor trail - which is the one case that should always be avoided, as it produces measurements of the vertical wind only, and consequently division by zero when trying to deduce "horizontal winds". I suggest using the figure, but tilt it at some arbitrary (non-zero!) angle from the horizontal. | Agreed, modified as suggested. |

| | | |
|---|---|---|
| 3. | 3. The paper does not discuss how the Bragg angles are found. The paper by Stober et al. (2018) suggests they always point exactly to the centre-point of the transmitter-receiver line, but this is in error, and only approximately true (as some simple geometric calculations using high-school-level trigonometry will show). I believe the authors of the Stober et al. paper sent a correction to the journal in this regard. Incorrect calculation of the Bragg angle will bias the results. | By "Bragg angle", we assume the reviewer is referring to the direction of the vector that is perpendicular to the meteor trail. In this paper, we have assumed this is in the direction of the bisector of the vectors from the meteor to the transmitter and the meteor to the receiver (which have been referred to as $t$ and $b$, respectively), and the way this has been estimated has already been discussed in Sect. 3.6. We agree with the reviewer's point that this vector does not necessarily point towards the midpoint between the transmitter and receiver, as suggested in *Stober* et al. (2018), Fig. 9. As part of this correction, we have made some modifications to how we discuss the use of the Bragg angle/vector in the section on wind (3.7) and covariance (3.9) estimation. |
| 4. | 4. The appendices (software) of Stober et al. contain couple of typos - the authors should confirm that they found these. | We found one typo in the appendices of this paper, and have now made note of this in the manuscript. |
| 5. | 5. The authors use a spherical-Earth calculation for the bistatic case, as they should. But it is unclear as to whether they did the same in the monostatic case. In some radars, a flat-Earth approximation is used for the monostatic case - the error in height is typically 0.5 km or so at zenith angles of 45 degrees. But it is unclear here whether the flat-Earth approximation was used in the monostatic case, or whether a full spherical Earth approximation was used in both cases. If a flat-earth approximation is used in one case, and a spherical-Earth system used in another, it could lead to biases in comparison. This may also impact the upper-right graph in Fig. 1 | We have added comments in several places to clarify that we assumed an ellipsoidal Earth for both bistatic and monostatic cases. |
| 6. | please add labels a,b,c,d - its getting annoying referring to "upper left", "lower right" etc. | Added labels to all figures where appropriate. |
| 7. | Another point of note, though maybe less important, is the mean Bragg wavelength. The authors note that the bistatic procedure gives rise to a diversity of Bragg scales, but it must be work, so a shift to 47 MHz might be more dramatic (relatively). . . . It would be of interest for the authors to look not just at diversity effects but also to consider the mean effective wavelength due to the bistatic arrangement. | This is a good point—we have added a histogram of the effective radar frequency to the paper. |

| | | |
|---|---|---|
| 8. | Page 6, line 10 - the term "said distributions" seems clumsy - the word "said" is used in this way several times - it does not add anything to the text, and I would recommend removing it. | Removed. |
| 9. | Page 8, line 4 - the authors talk about "well-known linear GW polarization relations". There is no single set of "linear relations" -this is just one of many, and it is the simplest. It all depends on whether you include Coriolis terms, Brunt-Vaisala terms, scale-height terms etc -these cases still deal with linear waves, but the detailed formulas change. . . . So this misleading statement needs to be clarified, perhaps by saying "we use the simplest polarization relations". Of course it needs to be recognized that the authors own work will be in error at periods close to the tidal and inertial frequencies if they use this simple polarization relation - Coriolis terms should be included for a proper treatment, for example. This is pertinent to discussions later in the text, where the authors consider the optimum bandwidth for studies and look at GW periods close to the tides. | Thank you for raising this important point. We have removed the "well-known" appellation. In addition, we only use wave periods up to 4 hours in the simulations, which is well below the inertial and predominant tidal periods. Therefore, we believe it is unnecessary to qualify the potential lack of realism associated with using the simpler polarization relations. |
| 10. | Page 9, line 9 — I would make the "dot product" between vecef and e clearer - maybe use a bold dot - it is important that readers do not miss this, and plays into the discussions introduced earlier regarding overhead meteors. The equation emphasizes that when the meteors are close to overhead, the radial velocity is dominated by the vertical wind component | Agreed, done. |
| 11. | Incidentally, the authors used bold font to represent both vectors and matrices. While not wrong, since a vector is indeed a type of matrix, they may want to consider distinguishing these cases e.g. by using an arrow over the vectors and keeping matrices bold. | We have tried doing this, but the Copernicus LaTeX class does not permit arrows over vectors. All vectors have been changed to bold/italicized to distinguish them from matrices (bold). |
| 12. | Page 10, line 18 - how "rare" is "rare"? | Thank you for picking up on this—we have re-examined and clarified accordingly. |

| | | |
|---|---|---|
| 13. | Page 10, lines 19 to 21, and page 11, lines 1 - 14. These lines are very disturbing. First, the authors seem to have the wrong Holdsworth et al. (2004) paper in the References. The authors give Holdsworth, D. A., Tsutsumi, M., Reid, I. M., Nakamura, T., and Tsuda, T.: Interferometric meteor radar phase calibration using meteor echoes, Radio Sci., 39, 2004. Yet line 19 on page 10 discusses an "error code 3". I cannot see an "error code 3" in this paper. I CAN find one in the paper "Holdsworth, D. A., I. M. Reid, and M. A. Cervera (2004), Buckland Park all-sky interferometric meteor radar, Radio Sci., 39, RS5009, doi:10.1029/2003RS003014". It appears that the authors have referenced the wrong paper, and I believe this is true for (almost?) all references to Holdsworth et al., (2004). | Thank you for picking up on the incorrect reference—this was a mistake on our part. This has been corrected. |
| 14. | But it gets worse. On page 11, line 3, the authors give Holdsworth (2004) as a representative example of "standard analysis", and then on line 11, make the statement "... we follow the iterative scheme PROPOSED by Holdsworth et al., (2004).". Sorry ,but this is very misleading. This iterative scheme was proposed fully 7 years earlier (Hocking and Thayaparan, Radio Sci., 32, 833-865, 1997, section 4.1) and again discussed in Hocking et al., J. Atmos. Solar-Terr. Physics, 63, 155-169, 2001.). Holdsworth may have made minor adjustments, but he did not "propose" the idea - it was already proposed 7 years earlier. Indeed much of Table 2 in Holdsworth et al (2004) was also proposed 3 years before Holdsworth (2004) - in Hocking et al., (2001). Repeated reference to Holdsworth throughout the text in regard to ideas which were proposed many years earlier by others is unprofessional and misleading. | We apologize for missing this reference—this has now been added. |
| 15. | In regard to sections 3.7 and on - I again ask - were cases close to the zenith, where the radial velocity is dominated by vertical winds, removed, or given lower weighting? Or how were such cases dealt with? Nasty near-infinity terms can occur when such terms are include, which could bias the resultant mean winds and momentum fluxes. | As discussed earlier in this reply, we did not consider removing/weighting the meteors in this region given the small number of detections here. Again, we believe that weighting the meteor distribution to minimize the wind/momentum flux estimation errors is a separate topic warranting its own investigation, and have mentioned this in the discussion section of the manuscript. |

| | | |
|---|---|---|
| 16. | Page 12, lines 1 to 9. This is another example of poor citations, perhaps due to the first author's inexperience. Line 1 says "The approach we apply is identical to that presented by Thorsen et al., (1997) and subsequently Hocking (2005)." It gives the impression that the ideas were developed by Thorsen and simply re-applied by Hocking. This is patently untrue. The current authors use the matrix A' as discussed in lines 7 to 8. Thorsen et al. NEVER used this matrix. The matrix Thorsen et al. used is given in their equation (18), section 3.3, page 714. It is a MUCH simplified version of the true A'. The matrix that the current authors use comes straight out of Hocking (2004) and was never used by Thorsen et al. It makes a major improvement to the analysis. So please replace "subsequently" in line 1 of page 1 with "subsequently improved", and give a specific reference to Hocking (2004) in regard to lines 7 and 8. | We have adjusted this section to avoid implying that Hocking (2005) simply re-applied the estimator shown in Thorsen et al. (1997). However, it should be noted that the two approaches are fundamentally the same; the Hocking (2005) paper is just more descriptive in that it outlines a way of inverting the A' matrix (by directly finding the least squares solution to the problem). We also note in this paper that we do not use the inversion method described by Hocking (2005), but rather SVD, due to the fact that SVD is known to provide more reliable solutions to poorly-conditioned problems. |
| 17. | Page 12, line 26 - "we have opted to compute the covariances at the origin" - do the authors mean "at the origin at the ground" or "at a point in the atmosphere at meteor heights immediately above the ground". I can see that either might work, but it needs to be clarified. | This has now been clarified. |
| 18. | Page 13, line 1. The statement that "This estimate represents what one would measure with an "anemometer" at some fixed location in the vicinity of the radar", is a little misleading. I understand what the authors are trying to say, but it isn't quite right, and needs some caveats added. First, "in the vicinity of the radar" is misleading. Do the authors mean "in the vicinity of the meteor region immediately above the radar"? As it reads, it could be at ground-level. In addition, in the bistatic case, which radar? the transmitter? or the receivers? True, in this case they are moderately close, but since the report is supposed to somehow represent general multistatic radars, where the transmitter and receiver can be hundreds of km apart, the question is no longer pedantic. | We have removed the statements about the anemometer and about the reference being "in the vicinity of the radar". We have already briefly discussed the point about the correlation of the wave field over the entire observation volume in Sect. 3.10; we found the covariances evaluated at the coordinate system origin vs. at the positions/times of the meteors to be in very close agreement. On the basis of this, we assumed use of the "origin" reference point to be valid. |

| | | |
|---|---|---|
| 19. | Furthermore, a suitable anenometer might truly measure wind components, where with the radar we have this problem that we deal with radial components. So while I get what the author is trying to say, I suggest it be mentioned with some more caution than has been applied.. | I do not understand the specific problem the reviewer is discussing here. In any case, the whole premise of the simulation has been to estimate the errors associated with propagating realistic radial velocity errors to the covariance estimates; the "perfect anemometer" winds, assuming the radar is attempting to measure these, are used as the reference for estimating the size of the error. |
| 20. | Page 14, lines 9-10 - OK, but as discussed earlier, could some of the increase in count rates be due to a new effective mean frequency? And I also ask again - did the monostatic radar use the same "spherical Earth" geometry as the bistatic case? | Thank you for picking up on the lack of clarity here—we have reworded this sentence to make it clear that we are talking about the increase in counts associated with combining the monostatic and bistatic receiver data. There is indeed a slight increase in counts at the higher altitudes due to the effective lower operating frequency, but this is not related to the point being made, and has already been mentioned in Sect 2.1. As discussed earlier in this reply, we also used the same ellipsoidal Earth model to compute meteor-related parameters in the monostatic and bistatic cases. |
| 21. | Pages 15-21 seem quite good. However, the term "gap" is used in Figs. 10, 11, in line 10, page 21. "Gap" gives the impression of a "hole" - I would say the term "temporal shift" might be more appropriate. | Agreed, made suggested change. |
| 22. | Page 21, line 1 - the authors say ".. aside from considering that the GWs may have propagated from a region with weak eastward mesospheric winds." Do they really mean this? Or do they mean "... GWs may have propagated through a region with weak eastward..."? It's the filtering that matters - so either the statement needs some expansion, or the authors really mean "through". | Thank you for picking up on this—changed to "through" as suggested. |
| 23. | Page 23 line 2 - give a reference to the forcing formula. | Done. |
| 24. | Page 23, line 28 - suggest changing "Also like.." to "As for..." | Done. |
| 25. | Page 24, lines 11-12 - seems a bit unclear - please try to reword what you mean. | Thank you for picking up on this lack of clarity—modified. |
| 26. | Page 28, conclusions. Some of these statements may need revisiting following other comments discussed here-in. | None of the modifications affect the conclusion section, so we have left it unchanged. |

[revised manuscript text omitted]